# DYNAMIC ASSORTMENT SELECTION AND PRICING WITH LEARNING

## ABSTRACT

We consider a dynamic assortment selection and pricing problem in which a seller has $n$ different items available for sale. In each round, the seller observes $d$-dimensional contextual preference information for the user and offers to the user an assortment of $K$ items at prices chosen by the seller. The user selects at most one of the products from the offered assortment according to a multinomial logit choice model whose parameters are unknown. The seller observes which, if any, item is chosen at the end of each round, with a goal of maximizing cumulative revenue over a selling horizon of length $T$. For this problem, we propose an algorithm that learns from user feedback and achieves $n$-independent revenue regret of order $\widetilde{\mathcal{O}}(d\sqrt{T})$. We also show that this regret rate is optimal, up to logarithmic factors, by obtaining lower bounds for the regret achievable by any algorithm.

## 1 INTRODUCTION

In online marketplaces, dynamic assortment selection and pricing for sequentially arriving buyers presents a challenging context for online learning. Since the preferences of buyers are varying and uncertain, adaptive strategies are essential to meet their needs and maximize the effectiveness of offers. To address this problem, we investigate the application of online learning techniques for contextual assortment selection and pricing. Assortment selection involves the seller choosing a subset of items from a vast catalog to present to buyers, and dynamically assigning prices to the offered items. The overall goal is to maximize revenue over the course of repeated interactions.

Dynamic assortment selection and pricing strategies are deployed in a variety of online sectors including e-commerce (e.g., Amazon, eBay), travel (e.g., Expedia), hospitality (e.g., Airbnb), and food delivery (e.g., Doordash). With similar systems becoming ubiquitous in our daily lives, there is a growing opportunity to deliver tailored product recommendations and pricing adjustments. Therefore, it is crucial to consider data-driven approaches that can enhance user experiences and boost profitability in today's highly competitive digital industry.

We consider designing sequential *assortment selection* and *pricing* algorithms that offer a sequence of menus (assortments) of up to $K$ items from a catalog (set) of $n$ possible items. The learning agent (seller) makes sequential decisions and receives human (user) feedback. The feedback at each round is the particular item chosen by the user from the offered assortment. We assume that the item choice follows a multinomial logistic (MNL) model (McFadden, 1978), which is one of the most widely used models in dynamic assortment optimization literature (Caro & Gallien, 2007; Rusmevichientong et al., 2010; Sauré & Zeevi, 2013; Agrawal et al., 2017; Aouad et al., 2018; Agrawal et al., 2019). Because assortment-based offers are relevant to many industries that involve access to additional information about users, contextual assortment selection models have gained significant traction in recent years (Chen et al., 2020; Oh & Iyengar, 2021). In alignment with this approach, we assume that the utility parameters in the MNL choice model are linear functions of $d$-dimensional context vectors that are revealed at each round.

To faithfully address a range of real-world scenarios where price optimization is essential for maximal revenue, we incorporate the pricing of items in the offered assortment as a second component of the seller's problem. This differs from most, if not all, previous literature on sequential assortment selection, wherein prices (or revenues) are assumed to be predetermined for each item in the catalog.

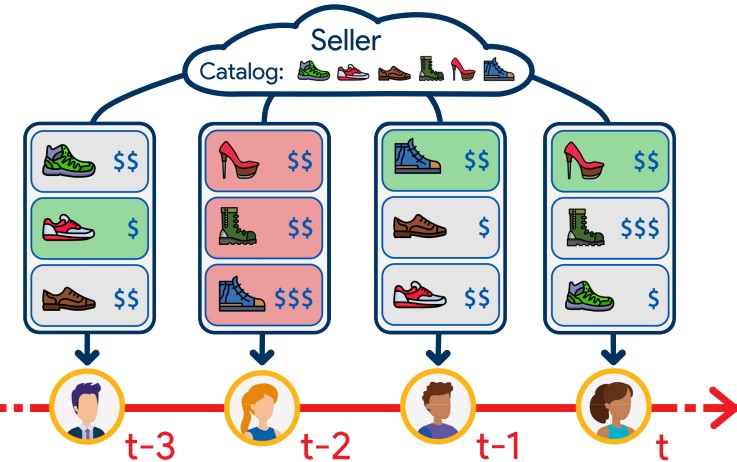

Figure 1: A seller has access to a set (catalog) of $n = 6$ distinct items, from which it can advertise to sequentially arriving users. In each round, the seller offers an assortment of $K = 3$ items at well-chosen prices. The user selects one of the products from the offered assortment (represented with a green background), or rejects all offered items (represented with a red background).

In the process of offering a sequence of assortments with judiciously chosen prices, the seller's goal is to maximize the expected revenue accumulated over a time horizon of $T$ rounds. However, since the seller does not have knowledge of the parameters of the contextual choice model ahead of time, it encounters the dilemma of exploration vs. exploitation. In particular, the seller's decisions involve a trade-off between learning the choice model in order to increase long-term revenues and earning short-term revenues by leveraging the already-acquired information.

We tackle this challenge by developing an upper confidence bound (UCB) based algorithm that computes tight upper bounds for the utility parameters in the MNL model. Then, using these upper bounds, it calculates *optimistic* allocations and pricing vectors that strike a balance between exploration and exploitation. Consistent with the sequential decision-making literature, we measure the performance of algorithms using a relevant notion of regret, defined as the difference between the expected revenue generated by the algorithm and the offline optimal expected revenue when all parameters are known. We show that our algorithm enjoys a revenue regret of order $\widetilde{\mathcal{O}}(d\sqrt{T})$, which, as we show, is the best possible up to logarithmic factors in $d$ and $T$.

## 1.1 RELATED WORKS

**Generalized Linear Bandits:** For sequential decision-making with contextual information, linear bandits, generalized linear bandits, and their variants have been widely studied (Rusmevichientong & Tsitsiklis, 2010; Abbasi-Yadkori, 2011; Chu et al., 2011; Li et al., 2017). Nonetheless, these methods are limited to modeling the single-item selection scenario, which is becoming less common in practice compared to the multiple-item offering scenarios that we focus on in this work. There is a line of literature that considers combinatorial variants of the contextual bandit problem mostly with semi-bandit or cascading feedback (Chen et al., 2013; Qin et al., 2014; Kveton et al., 2015; Zong et al., 2016). However, these methods fail to capture the substitution effects since they do not take the user choice model into account. In contrast, the item choice (feedback) that we consider under the multinomial logit (MNL) model is a function of all items in the offered assortment as well as their prices.

**MNL Bandits:** There has been an emerging body of literature on multinomial logit (MNL) bandits in both non-contextual (Cheung & Simchi-Levi, 2017; Agrawal et al., 2017; 2019) and contextual settings (Chen et al., 2020; Oh & Iyengar, 2021). While these studies address the sequential assortment selection problem under the MNL choice model, their algorithms exclusively operate based on the assumption of fixed prices (or revenues) for the items. Consequently, they do not account for the potential effects of price optimization strategies that could be employed when presenting items to consumers.

Table 1: Comparison of related works and provided regret bounds. $T$ is the number of rounds, $K$ is the assortment size, $n$ is the total number of items, $d$ is the feature dimension. The big-$\mathcal{O}$ and big-$\Omega$ notations denote the regret upper and lower bounds, respectively. To the best of our knowledge, we are the first to jointly address the problems of contextual assortment selection and pricing.

| | **Context** | **Assortment** | **Pricing** | **Regret** |
|---|---|---|---|---|
| Agrawal et al. (2019) | No | Yes | No | $\widetilde{\mathcal{O}}(\sqrt{nT}), \Omega(\sqrt{nT/K})$ |
| Chen et al. (2020) | Yes | Yes | No | $\widetilde{\mathcal{O}}(d\sqrt{T}), \Omega(d\sqrt{T}/K)$ |
| Oh & Iyengar (2021) | Yes | Yes | No | $\widetilde{\mathcal{O}}(\sqrt{dT})$ |
| Javanmard et al. (2020) | Yes | No | Yes | $\mathcal{O}(\log(Td)\sqrt{T})$ |
| Perivier & Goyal (2022) | Yes | No | Yes | $\widetilde{\mathcal{O}}(d\sqrt{T})$ [1] |
| **This Work** (Algorithm 2) | Yes | Yes | Yes | $\widetilde{\mathcal{O}}(d\sqrt{T}), \Omega(d\sqrt{T})$ |

**Bandits for Dynamic Pricing:** The problem of dynamic pricing has been typically modeled as a variant of the multi-armed bandit problem that aims to maximize revenue from selling copies of a single good to sequentially arriving users (Kleinberg & Leighton, 2003; Besbes & Zeevi, 2009; Bubeck et al., 2019; Paes Leme & Schneider, 2018; Xu & Wang, 2021). However, all of these works address the pricing problem of offering a single item in each round. Our contribution stands out by considering the combinatorial aspect of the assortment selection problem faced in simultaneously offering multiple items, a factor that was not taken into account in most of the prior literature on dynamic pricing. A recent study by Javanmard et al. (2020) considers the problem of pricing multiple items that are offered under the MNL choice model. However, in contrast to our work, their framework assumes that all available items are offered to the buyer and hence the seller does not need to decide on an assortment as a part of its actions. In their work, they propose an almost-optimal algorithm that can achieve $\mathcal{O}(\log(Td)\sqrt{T})$ regret for their pricing-only setting. Comparing this result with the regret lower bound of order $\Omega(d\sqrt{T})$ for our problem, we see that the problem of dynamic assortment selection and pricing is fundamentally harder. Another recent study by Perivier & Goyal (2022) also considers the problem of pricing multiple items that are offered under the MNL choice model with the additional assumption of an adversarial arrival model for users. Similarly, they also do not consider assortment selection decisions while optimizing the prices of the items.

**Reinforcement Learning with Human Feedback:** The framework of reinforcement learning with human feedback (RLHF) has recently gained popularity through its empirical success in aligning human values with machine learning systems, including InstructGPT (Ouyang et al., 2022). The central goal of RLHF is to learn the rewards of different actions using human feedback that is received in the form of pairwise or $K$-wise comparisons between actions. Notably, in many deployments of RLHF, the human feedback is modeled through the Plackett-Luce (PL) model which is equivalent to the multinomial logit (MNL) choice model that we employ in our analysis (Luce, 2012; Liu, 2009; Christiano et al., 2017; Ouyang et al., 2022; Zhu et al., 2023).

## 1.2 OUR CONTRIBUTIONS

To the best of our knowledge, we are the first to address the problem of dynamic contextual assortment selection and pricing. Our contributions are as follows:

- We introduce and formalize the problem of sequential assortment selection and pricing under contextual MNL choice probabilities.

- We develop an upper confidence bound (UCB) based algorithm for the contextual assortment selection and pricing problem (Algorithm 2). We show that it achieves $n$-independent $\widetilde{O}(d\sqrt{T})$ regret in $T$ rounds where $d$ is the dimension of the context vectors.

- We further improve the time and space complexity of our algorithm by leveraging online Newton step (ONS) techniques for parameter estimation.

- We show that for any algorithm, there exists an adversarial problem instance such that it incurs $\Omega(d\sqrt{T})$ regret. Therefore, Algorithm 2 enjoys optimal regret up to logarithmic terms in $d$ and $T$.

---

[1] Their work considers an adversarial arrival model.

## 2 PROBLEM DEFINITION

**Notation:** We use bold lowercase font for vectors $\boldsymbol{x}$ and uppercase font for matrices $X$. For a vector $\boldsymbol{x}$, we denote its $i$-th entry by $x_i$ and we use $\|\boldsymbol{x}\|$ to denote its $\ell^2$-norm. For two vectors $\boldsymbol{x}$ and $\boldsymbol{y}$, we use $(\boldsymbol{x};\boldsymbol{y})$ to denote their concatenation and use $\langle\boldsymbol{x},\boldsymbol{y}\rangle$ to denote their inner product. For a vector $\boldsymbol{x}$ and a positive semi-definite matrix $W$, we use $\|\boldsymbol{x}\|_W$ to denote the weighted $\ell^2$-norm. For any positive integer $n$, we use $[n]$ to denote the set $\{1, 2, \ldots, n\}$.

We consider the problem of online assortment selection and pricing for selling items to sequentially arriving buyers. We denote the set of available items by $[n]$ and consider that the seller is constrained to offer at most $K$ items to each buyer. Accordingly, we let $\mathcal{S}_K := \{S \subseteq [n] : |S| \leq K\}$ denote the set of all possible assortments that the seller can choose to offer.

At each time $t \in [T]$, the seller observes random feature vectors $\boldsymbol{x}_{ti} \in \mathbb{R}^d$ for each item $i \in [n]$. Given this contextual information, the seller offers an assortment of items $S_t \in \mathcal{S}_K$ and chooses a price $p_{ti} \in \mathbb{R}_+$ for each offered item $i \in S_t$. At the end of each round $t$, the seller observes only the purchase decision $i_t \in S_t \cup \{0\}$ of the buyer and obtains revenue $p_{ti_t}$. Here, $\{0\}$ represents the no-purchase option (or outside option), which indicates that the user did not choose any item offered in $S_t$, resulting in revenue $p_{t0} = 0$.

For convenience, we let $\boldsymbol{p}_t \in \mathbb{R}_+^n$ denote the collection of prices chosen for all items where the prices are set to $p_{ti} = 0$ for items that are not offered, i.e. $i \notin S_t$.

For a given assortment $S_t$ and price vector $\boldsymbol{p}_t$, the buyer's decision $i_t$ is a categorical random variable with support $S_t \cup \{0\}$. We model this decision via the widely used multinomial logit (MNL) choice model (McFadden, 1978) under a linear utility function. Formally, the choice probability for each item $i \in S_t$ (and the no-purchase option) is assumed to be given as in the following assumption.

**Assumption 1** (Multinomial logit choice under linear utility)**.** *The utility of the buyer at time $t$ for item $i$ is given by the linear model*

$$u_{ti}(p) = \langle \boldsymbol{\psi}^*, \boldsymbol{x}_{ti}\rangle - \langle \boldsymbol{\phi}^*, \boldsymbol{x}_{ti}\rangle \cdot p$$

*where $\boldsymbol{\psi}^* \in \mathbb{R}^d$ and $\boldsymbol{\phi}^* \in \mathbb{R}^d$ are time-invariant parameter vectors unknown to the seller agent. In this model, the $\alpha_{ti} := \langle\boldsymbol{\psi}^*, \boldsymbol{x}_{ti}\rangle$ term represents the buyer's base valuation of the item while the $\beta_{ti} := \langle\boldsymbol{\phi}^*, \boldsymbol{x}_{ti}\rangle$ term represents the buyer's price sensitivity.*

*Then, given an assortment $S_t$ with prices $\boldsymbol{p}_t$, the probability that the buyer selects item $i \in S_t$ is*

$$q_t(i|S_t, \boldsymbol{p}_t) := \frac{\exp\{u_{ti}(p_{ti})\}}{1 + \sum_{j \in S_t} \exp\{u_{tj}(p_{tj})\}}, \quad i \in S_t.$$

*Consequently, the probability of no purchase is*

$$q_t(0|S_t, \boldsymbol{p}_t) := \frac{1}{1 + \sum_{j \in S_t} \exp\{u_{tj}(p_{tj})\}}.$$

Under the MNL model, the expected revenue at time $t$ is given by

$$R_t(S_t, \boldsymbol{p}_t) := \sum_{i \in S_t} p_{ti} \cdot q_t(i|S_t, \boldsymbol{p}_t) \tag{1}$$

for any selection of assortment $S_t \in \mathcal{S}_K$ and price vector $\boldsymbol{p}_t \in \mathbb{R}_+^n$. Thus, for a sequence of assortments $S_t \in \mathcal{S}_K$ and price vectors $\boldsymbol{p}_t \in \mathbb{R}_+^n$ chosen for each time $t \in [T]$, the cumulative expected revenue can be written as $\sum_{t=1}^{T} R_t(S_t, \boldsymbol{p}_t)$.

After the seller decides on the assortment $S_t \in \mathcal{S}_K$ and prices $\boldsymbol{p}_t \in \mathbb{R}_+^n$ to offer to the user at each time $t$, the user reports the item $i_t \in S_t \cup \{0\}$ that they have decided to purchase. We denote by $H_t$ the history $\{\{\boldsymbol{x}_{\tau i}\}_{i \in [n]}, S_\tau, \boldsymbol{p}_\tau, i_\tau\}_{\tau=1}^{t-1}$ of observations available to the seller when choosing the next set of assortment $S_t \in \mathcal{S}_K$ along with the next price vector $\boldsymbol{p}_t$. Then, the seller agent employs a policy $\boldsymbol{\pi} = \{\pi^t | t \in [T]\}$, which is a sequence of functions, each mapping the history $H_t$ and the context vectors $\{\boldsymbol{x}_{ti}\}_{i \in [n]}$ to an action $(S_t, \boldsymbol{p}_t) \in \mathcal{S}_K \times \mathbb{R}_+^n$.

Given the contextual information at every round $t$, the task of the seller is to sequentially offer the items to users at well-chosen prices such that it can achieve maximal revenue. To evaluate policies in achieving this objective, we define the *regret* metric that measures the gap between the expected revenue of policy $\boldsymbol{\pi}$ and that of the offline optimal sequence of assortments and prices. The regret $\mathcal{R}_T$ for a time horizon of $T$ periods is defined as

$$\mathcal{R}_T := \sum_{t=1}^{T} R_t(S_t^*, \boldsymbol{p}_t^*) - \sum_{t=1}^{T} R_t(S_t, \boldsymbol{p}_t),$$

where $(S_t^*, \boldsymbol{p}_t^*)$ denotes an offline optimal assortment and price selection that satisfies

$$(S_t^*, \boldsymbol{p}_t^*) \in \underset{\substack{S \in \mathcal{S}_K \\ \boldsymbol{p} \in \mathbb{R}_+^n}}{\arg\max} R_t(S, \boldsymbol{p}). \tag{2}$$

Based on this definition of the regret metric, we see that regret minimization is equivalent to maximizing the cumulative expected revenue.

## 3  OPTIMAL ASSORTMENT SELECTION AND PRICING

As stated in Assumption 1, we assume that buyers' purchase decisions are given by a multinomial logit (MNL) model. Therefore, the assortment and price optimization at time $t$ can be formulated as

$$\max_{\substack{S_t \in \mathcal{S}_K \\ \boldsymbol{p}_t \in \mathbb{R}_+^n}} R_t(S_t, \boldsymbol{p}_t) = \max_{\substack{S_t \in \mathcal{S}_K \\ \boldsymbol{p}_t \in \mathbb{R}_+^n}} \sum_{i \in S_t} p_{ti} \cdot q_t(i|S_t, \boldsymbol{p}_t) \tag{3}$$

$$= \max_{\substack{S_t \in \mathcal{S}_K \\ \boldsymbol{p}_t \in \mathbb{R}_+^n}} \frac{\sum_{i \in S_t} p_{ti} \exp\{u_{ti}(p_{ti})\}}{1 + \sum_{j \in S_t} \exp\{u_{tj}(p_{tj})\}}. \tag{4}$$

We also recall that the utility functions are given by linear form $u_{ti}(p) = \alpha_{ti} - \beta_{ti} p$ where $\alpha_{ti} = \langle \boldsymbol{\psi}^*, \boldsymbol{x}_{ti} \rangle$ and $\beta_{ti} = \langle \boldsymbol{\phi}^*, \boldsymbol{x}_{ti} \rangle$.

Next, we make the following regularity assumption.

**Assumption 2.** *There exists a constant $L_0 > 0$ such that price sensitivity $\beta_{ti} = \langle \boldsymbol{\phi}^*, \boldsymbol{x}_{ti} \rangle$ satisfies $\beta_{ti} \geq L_0$ for all $t \in [T]$ and $i \in [n]$, almost surely.*

This assumption ensures that the utility function $u_{ti}(p)$ is decreasing in price and hence infinity is a so-called null price, i.e. $\lim_{p \to \infty} p e^{u_{ti}(p)} = 0$ for all $i \in [n]$. This property is crucial in ensuring that the objective function in equation 4 has a finite maximum. Because if we had $\langle \boldsymbol{\phi}^*, \boldsymbol{x}_{ti} \rangle \leq 0$ for some $i \in S_t$, we would have $\lim_{p \to \infty} p e^{u_{ti}(p)} = \infty$ and hence, letting $p_{ti} \to \infty$ would cause the objective function (i.e., expected revenue) to increase without bound. To avoid this complication, we make the regularity assumption given in Assumption 2.

Under the MNL choice model with known linear utility functions, Wang (2013) shows that the optimum assortment and prices can be characterized as in the following proposition.

**Proposition 1** (Optimum assortments and prices). *Under linear utility functions $u_{ti}(p) = \alpha_{ti} - \beta_{ti} p$ with $\beta_{ti} > 0$ for all $i \in [n]$, the optimum prices are given by*

$$p_{ti}^* = \frac{1}{\beta_{ti}} + B_t,$$

*where $B_t$ is defined to be the unique solution of the fixed point equation*

$$B = \max_{S \in \mathcal{S}_K} \sum_{i \in S} v_{ti}(B) \tag{5}$$

*for $v_{ti}(B) := e^{\alpha_{ti} - \beta_{ti} B - 1} / \beta_{ti}$. Furthermore, the optimum assortment $S_t^*$ is the assortment $S$ that achieves the maximum in the optimization problem in equation 5, and the optimum revenue achieved by $(S_t^*, \boldsymbol{p}_t^*)$ is equal to $B_t$.*

To solve the fixed point equation given in equation 5, we observe that $v_{ti}(B)$ is a strictly decreasing function in $B$ for any $i \in [n]$. Hence, we can show that the right-hand side of equation 5 is a strictly decreasing function in $B$, implying that it has a unique fixed point.

As we will show in Lemma 1, our regularity assumptions ensure that this fixed point lies in the interval $[0, P_0]$ for some finite $P_0$. Therefore, we can appeal to a binary search algorithm over the interval $[0, P_0]$ to find the fixed point. Since each iteration of this binary search algorithm requires us to compute $v_{ti}(B)$ value for all $i \in [n]$, it has a computational complexity of $\mathcal{O}(n)$. For the sake of completeness and future reference, we describe this procedure in Algorithm 1.

---

**Algorithm 1** Assortment selection and pricing for linear utility functions

---

1: **Input:** Accuracy parameter $\epsilon$, search interval $[0, P_0]$, utility parameters $\alpha_{ti}$ and $\beta_{ti}$ for $i \in [n]$
2: $B_\ell = 0$, $B_r = P_0$
3: **while** $B_r - B_\ell > \epsilon$ **do**
4: $\quad B \leftarrow (B_r + B_\ell)/2$
5: $\quad$ **if** $B > \max_{S \in \mathcal{S}_K} \sum_{i \in S} v_{ti}(B)$ **then** $B_r \leftarrow B$ **else** $B_\ell \leftarrow B$
6: **Output:** $B$

---

## 4 METHODOLOGY

In this section, we discuss how to estimate parameters based on user choices, introduce our assortment selection and pricing algorithms, and provide their regret bounds.

### 4.1 MLE FOR MULTINOMIAL LOGISTIC REGRESSION

Since the seller does not have access to problem parameters $\boldsymbol{\psi}^* \in \mathbb{R}^d$ and $\boldsymbol{\phi}^* \in \mathbb{R}^d$, it cannot directly compute the optimum assortments and prices given by Proposition 1. Therefore, it needs to construct an estimate of the parameters based on the history $H_t$ of observations. In this work, we consider a policy that uses the maximum likelihood estimate (MLE) of the parameters as we briefly describe in this section.

For convenience, we let $\boldsymbol{\theta} = (\boldsymbol{\psi}; \boldsymbol{\phi})$ and $\widetilde{\boldsymbol{x}}_{ti} = (\boldsymbol{x}_{ti}; -p_{ti}\boldsymbol{x}_{ti})$ denote the extended parameter and feature vectors such that

$$\langle \boldsymbol{\theta}, \widetilde{\boldsymbol{x}}_{ti} \rangle = \langle \boldsymbol{\psi}, \boldsymbol{x}_{ti} \rangle - \langle \boldsymbol{\phi}, \boldsymbol{x}_{ti} \rangle \cdot p_{ti}.$$

Then, we write the MNL choice probabilities under parameter $\boldsymbol{\theta} = (\boldsymbol{\psi}; \boldsymbol{\phi})$ using the notation

$$q_t(i|S_t, \boldsymbol{p}_t; \boldsymbol{\theta}) = \frac{\exp\{\langle \boldsymbol{\psi}, \boldsymbol{x}_{ti} \rangle - \langle \boldsymbol{\phi}, \boldsymbol{x}_{ti} \rangle \cdot p_{ti}\}}{1 + \sum_{j \in S_t} \exp\{\langle \boldsymbol{\psi}, \boldsymbol{x}_{tj} \rangle - \langle \boldsymbol{\phi}, \boldsymbol{x}_{tj} \rangle \cdot p_{tj}\}} = \frac{e^{\langle \boldsymbol{\theta}, \widetilde{\boldsymbol{x}}_{ti} \rangle}}{1 + \sum_{j \in S_t} e^{\langle \boldsymbol{\theta}, \widetilde{\boldsymbol{x}}_{\tau j} \rangle}}.$$

Based on the observations up to time $t$, the negative log-likelihood function is given by

$$\ell_t(\boldsymbol{\theta}) := -\sum_{\tau=1}^{t-1} \log q_\tau(i_\tau | S_\tau, \boldsymbol{p}_\tau; \boldsymbol{\theta}),$$

which is also known as the cross-entropy error function for the multi-class classification problem. Then, as we formalize in the next proposition, the maximum likelihood estimate is given as the minimizer of the negative log-likelihood function.

**Proposition 2.** *The maximum likelihood estimator is any parameter $\widehat{\boldsymbol{\theta}}_t$ that minimizes the negative log-likelihood function over the parameter space, that is*

$$\widehat{\boldsymbol{\theta}}_t \in \arg\min_{\boldsymbol{\theta}} \ell_t(\boldsymbol{\theta}). \tag{6}$$

*The negative log-likelihood function $\ell_t(\boldsymbol{\theta})$ is convex over $\boldsymbol{\theta} \in \mathbb{R}^{2d}$. Therefore, any parameter $\widehat{\boldsymbol{\theta}}_t$ that satisfies the first order optimality condition $\nabla_{\boldsymbol{\theta}} \ell_t(\widehat{\boldsymbol{\theta}}_t) = 0$ is a maximum likelihood estimate.*

*Furthermore, if the Gram matrix $V_{t-1} = \sum_{\tau=1}^{t-1} \sum_{i \in S_\tau} \widetilde{\boldsymbol{x}}_{\tau i} \widetilde{\boldsymbol{x}}_{\tau i}^\top$ is positive definite, then $\ell_t(\boldsymbol{\theta})$ is strongly convex and thus admits a unique minimizer.*

Since the negative log-likelihood function is convex over $\boldsymbol{\theta} \in \mathbb{R}^{2d}$, we can use gradient-based convex optimization methods to find an MLE solution (Boyd & Vandenberghe, 2004).

---

**Algorithm 2** DASP-MNL: Dynamic Assortment Selection and Pricing under MNL Model

---

1: **Input:** initialization rounds $T_0$, confidence parameters $\{\alpha_t\}_{t \in [T]}$, minimum price sensitivity $L_0$
2: $V_0 \leftarrow \mathbf{0} \in \mathbb{R}^{2d \times 2d}$
3: **for** $t = 1, 2, \ldots, T_0$ **do**                                                           ▷ initialization rounds
4:     Choose $S_t$ uniformly at random from $\{S \subseteq [n] : |S| \le K\}$
5:     Choose $p_{ti}$ independently and uniformly at random from $[1, 2]$ for all $i \in S_t$
6:     Offer assortment $S_t$ at price $\boldsymbol{p}_t$ and observe $i_t$
7:     $V_t \leftarrow V_{t-1} + \sum_{i \in S_t} \widetilde{\boldsymbol{x}}_{ti} \widetilde{\boldsymbol{x}}_{ti}^\top$
8: **for** $t = T_0 + 1, T_0 + 2, \ldots, T$ **do**
9:     Compute $g_{ti} := \alpha_t \|(\boldsymbol{x}_{ti}, \boldsymbol{x}_{ti})\|_{V_t^{-1}}$ for all $i \in [n]$                    ▷ Confidence bonus
10:     Compute $\widehat{\boldsymbol{\theta}}_t = (\widehat{\boldsymbol{\psi}}_t, \widehat{\boldsymbol{\phi}}_t)$ by solving equation 6                                   ▷ MLE
11:     Let $h_{ti}(p) = \min\{\langle \widehat{\boldsymbol{\psi}}_t, \boldsymbol{x}_{ti} \rangle + g_{ti}, 1\} - \max\{\langle \widehat{\boldsymbol{\phi}}_t, \boldsymbol{x}_{ti} \rangle - g_{ti}, L_0\} \cdot p$ for all $i \in [n]$
12:     Choose $(S_t, \boldsymbol{p}_t)$ using Algorithm 1 with linear functions $h_{ti}(p)$
13:     Offer assortment $S_t$ at price $\boldsymbol{p}_t$ and observe $i_t$
14:     $V_t \leftarrow V_{t-1} + \sum_{i \in S_t} \widetilde{\boldsymbol{x}}_{ti} \widetilde{\boldsymbol{x}}_{ti}^\top$

---

## 4.2   ALGORITHM

The basic idea of our algorithm is to construct an upper confidence bound for the revenue function $R_t(S, \boldsymbol{p})$. The upper confidence bound (UCB) techniques and the optimism in the face of uncertainty (OFU) principle have been widely known to be effective in balancing the exploration and exploitation in many bandit problems, including multi-armed bandits (Lattimore & Szepesvári, 2020), linear bandits (Dani et al., 2008; Abbasi-Yadkori, 2011) and generalized linear bandits (Li et al., 2017).

At each round $t$, our algorithm determines the assortments and prices according to the OFU principle in order to ensure low regret. In particular, we construct a pointwise confidence upper bound $h_{ti}(p)$ for each utility function $u_{ti}(p)$, i.e., $h_{ti}(p) \ge u_{ti}(p)$ for all $p \in \mathbb{R}_+$ with high probability.

In this construction, we use the maximum likelihood estimate $\widehat{\boldsymbol{\theta}}_t = (\widehat{\boldsymbol{\psi}}_t, \widehat{\boldsymbol{\phi}}_t)$ calculated by solving the maximum likelihood problem described above in equation 6. Then, given the MLE of the parameters, the obtained upper bound is of the form

$$h_{ti}(p) = \min\{\langle \widehat{\boldsymbol{\psi}}_t, \boldsymbol{x}_{ti} \rangle + g_{ti}, 1\} - \max\{\langle \widehat{\boldsymbol{\phi}}_t, \boldsymbol{x}_{ti} \rangle - g_{ti}, L_0\} \cdot p,$$

where $g_{ti} = \alpha_t \|(\boldsymbol{x}_{ti}, \boldsymbol{x}_{ti})\|_{V_t^{-1}}$ is the confidence bonus for some confidence radius $\alpha_t$. As a result, we can replace each $u_{ti}(p)$ in equation 1 with $h_{ti}(p)$ to obtain the revenue function upper bound

$$\widetilde{R}_t(S, \boldsymbol{p}) := \frac{\sum_{i \in S_t} p_{ti} \exp\{h_{ti}(p_{ti})\}}{1 + \sum_{j \in S_t} \exp\{h_{tj}(p_{tj})\}}. \tag{7}$$

As we verify in proving our regret bounds, this estimate satisfies $\widetilde{R}_t(S, \boldsymbol{p}) \ge R_t(S, \boldsymbol{p})$ for any $S \in \mathcal{S}_K$ and any $\boldsymbol{p} \in \mathbb{R}_+^n$. Using $\widetilde{R}_t$ as a proxy for $R_t$, we choose the assortments and prices according to

$$(S_t, \boldsymbol{p}_t) \in \underset{\substack{S \in \mathcal{S}_K \\ \boldsymbol{p} \in \mathbb{R}_+^n}}{\arg\max} \widetilde{R}_t(S, \boldsymbol{p}). \tag{8}$$

As discussed in Section 3, we can solve this optimization problem using the binary search method described in Algorithm 1 with estimated linear functions $h_{ti}(p)$.

## 4.3   REGRET ANALYSIS

Our main result presented in Theorem 3 concerns the regret upper bound for Algorithm 2. We show this result under the following assumption on the context process which is a standard assumption made in the generalized linear bandit (Li et al., 2017) and MNL contextual bandit (Chen et al., 2020; Oh & Iyengar, 2021) literature.

**Assumption 3** (Stochastic and bounded features). *Each feature vector $\boldsymbol{x}_{ti}$ is an independent random variable with unknown distribution; they satisfy $\|\boldsymbol{x}_{ti}\| \leq 1$, and there exists a constant $\sigma_0 > 0$ such that $\mathbb{E}[\boldsymbol{x}_{ti}\boldsymbol{x}_{ti}^\top] \succcurlyeq \sigma_0 \boldsymbol{I}$. Furthermore, parameter vectors satisfy $\|(\boldsymbol{\theta}^*, \boldsymbol{\phi}^*)\| \leq 1$.*

Accordingly, we can demonstrate in Theorem 3 that Algorithm 2 enjoys $\widetilde{\mathcal{O}}(d\sqrt{T})$ regret bound in terms of key problem primitives $n$, $d$ and $T$. This regret rate is independent of the number of items $n$, and is thus applicable in settings with a very large number of candidate items. Even though our regret upper bound does not capture the dependency with respect to the assortment size parameter $K$, the maximum assortment size is typically small (i.e., $K = \mathcal{O}(1)$) in many real-world applications.

**Theorem 3.** *Suppose Assumptions 1, 2, and 3 hold and we run DASP-MNL (Algorithm 2) with confidence width $\alpha_t$ given in equation 12 and initialization length $T_0$ given in equation 10. Then, the expected regret for a sufficiently large time horizon $T$ is upper-bounded as*

$$\mathcal{R}_T \leq C_1 \cdot d\sqrt{T \log T \log(T/d)}$$

*for some constant $C_1$ independent of $n$, $d$ and $T$.*

*Proof.* (Sketch) In proving our regret bounds, we first show that the fixed point $B_t$ defined in Proposition 1 lies within $[0, P_0]$ for some $P_0$, allowing us to constrain our search for the fixed point into a bounded interval. This result also implies that the optimum prices $p_{ti}^*$ are bounded within $[1, P]$ for some $P$. Then, we show that $T_0 = \Theta(d + \log T)$ rounds of random initialization is enough to ensure that $\Theta(\sigma_0^{-2}(d + \log T + P^2))$ is invertible at the end of the initialization phase with high probability. Similar to Li et al. (2017) and Oh & Iyengar (2021), the independence assumption (Assumption 3) on the feature vectors $\boldsymbol{x}_{ti}$ is only needed to ensure that $V_{T_0}$ is invertible at the end of the initialization phase. We do not require this stochasticity assumption in the rest of the regret analysis. Therefore, after the random initialization period of the first $T_0$ rounds, the context vectors $\boldsymbol{x}_{ti}$ can even be chosen adversarially as long as their norms $\|\boldsymbol{x}_{ti}\|$ are bounded and they satisfy the minimum price sensitivity condition $\langle \boldsymbol{\phi}^*, \boldsymbol{x}_{ti} \rangle \geq L_0$.

In the next step, we show that the assortments $S_t$ and prices $\boldsymbol{p}_t$ chosen at any round $t$ give rise to a sufficiently large probability of selection for any item $i \in S_t$ under any parameter $\boldsymbol{\theta}$ sufficiently close to $\boldsymbol{\theta}^*$. This condition is central in showing that the maximum likelihood estimator is consistent and satisfies a finite-sample normality-type estimation error bound. Based on these error bounds, we construct optimistic utility estimate functions $h_{ti}(p)$ that provide tight upper bounds for the true utility functions $u_{ti}(p)$ with high probability. This result in turn implies that $\widetilde{R}_t(S, \boldsymbol{p}) \geq R_t(S, \boldsymbol{p})$ for any $S \in \mathcal{S}_K$ and any $\boldsymbol{p} \in \mathbb{R}_+^n$. Finally, we decompose the regret into two parts where some suitable defined *good* event holds (with high probability) and it does not. We defer the additional details to Appendix B. □

### 4.4 EXTENSION TO ONLINE PARAMETER UPDATE

Algorithm 2 is simple to implement and enjoys provable regret bounds as shown in Theorem 3. However, the computation of the MLE at each round of Algorithm 2 requires access to all feature vectors corresponding to previous assortments. To reduce both the time and space complexities of our algorithm and improve its efficiency, we can instead use an online parameter update rule. The online version presented as Algorithm 3 in Appendix C finds an approximate MLE solution only using the context vectors corresponding to the last assortment. To achieve this, we use the fact that the negative log-likelihood function is strongly convex after initialization and apply a variant of the online Newton step discussed in Hazan et al. (2014); Zhang et al. (2016); Oh & Iyengar (2021). We show that the modified algorithm still enjoys $\widetilde{\mathcal{O}}(d\sqrt{T})$ even with the online update.

**Theorem 4.** *Suppose Assumptions 1, 2, and 3 hold and we run DASP-MNL with online parameter updates (Algorithm 3) with confidence width $\alpha_t$ given in equation 18 and initialization length $T_0$ given in equation 10. Then, the expected regret for a sufficiently large time horizon $T$ is upper-bounded as*

$$\mathcal{R}_T \leq C_2 \cdot d\sqrt{T \log T \log(T/d)}$$

*for some constant $C_2$ independent of $n$, $d$ and $T$.*

*Proof.* See Appendix C for the proof. □

## 4.5 Regret Lower Bounds for Assortment Selection and Pricing Problem

In this section, we establish a regret lower bound of order $\Omega(d\sqrt{T})$ in terms of key problem primitives $n$, $d$, and $T$ for the problem of assortment selection and pricing under the contextual MNL choice model. This result demonstrates that the proposed algorithm DASP-MNL (Algorithm 2) and its online version (Algorithm 3) are optimal, up to logarithmic terms in $d$ and $T$.

**Theorem 5.** *There exists a universal constant $C_3$ such that for any maximum assortment size $K \geq 1$, any minimum price sensitivity $L_0 > 0$, any context dimension $d$ divisible by 4, and any policy $\boldsymbol{\pi}$, there exists a worst-case problem instance with $n = \Theta(K \cdot 2^d)$ items, bounded context vectors (i.e., $\|\boldsymbol{x}_{ti}\| \leq 1$ for all $i \in [n]$), and bounded feature vectors (i.e., $\|(\boldsymbol{\theta}^*; \boldsymbol{\phi}^*)\| \leq 1$) such that the regret of $\boldsymbol{\pi}$ is lower bounded by $C_3 \cdot d\sqrt{T}/L_0$.*

*Proof.* (Sketch) At a high level, we prove this theorem in three steps. In the first step, we construct an adversarial set of parameters and reduce the task of lower bounding the worst-case regret of any policy to lower bounding the Bayes risk over the constructed parameter set. In the second step, we use a counting argument similar to the one used in Chen & Wang (2018) and Chen et al. (2020) to provide an explicit lower bound on the Bayes risk of the constructed adversarial parameter set. Finally, we apply Pinsker's inequality to complete the proof. We defer the details of the proof to Appendix D. □

## 5 Numerical Experiments

In this section, we demonstrate the efficacy of our proposed algorithms: DASP-MNL presented in Algorithm 2 and its online version Algorithm 3. We numerically evaluate our algorithms over independently generated problem instances and provide our results in Figure 2. In each instance, we generate problem parameters $(\boldsymbol{\psi}^*; \boldsymbol{\phi}^*)$ and context vectors $\boldsymbol{x}_{ti}$ by sampling their entries from uniform distributions such that we satisfy Assumptions 2 and 3. See Appendix E for further details. For various assortment sizes $K$ and various numbers of feature dimensions $d$, we run 20 independent problem instances with $n = 100$ items.

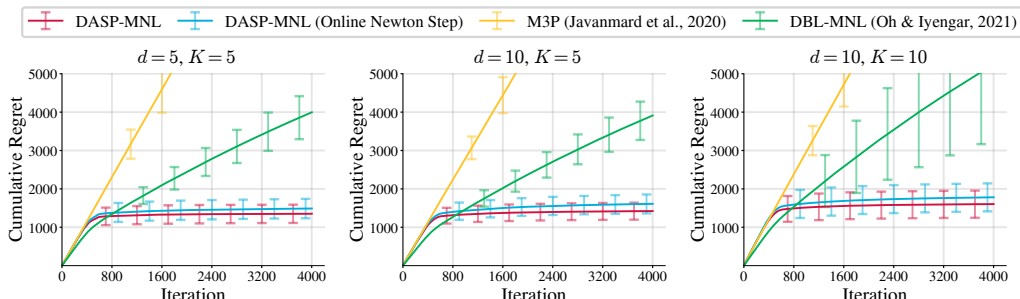

Figure 2: Cumulative regret of DASP-MNL (Algorithm 2), its online version (Algorithm 3), M3P (Javanmard et al., 2020), and DBL-MNL (Oh & Iyengar, 2021). The center lines show the mean across the runs while the error bars indicate two standard deviations. Results demonstrate the efficacy of our algorithms in achieving diminishing regret per round as our theoretical results predict.

We compare the performances of our proposed algorithms with those of a state-of-the-art MNL assortment selection algorithm DBL-MNL (Oh & Iyengar, 2021) and an MNL pricing algorithm M3P (Javanmard et al., 2020). Since DBL-MNL is only designed for assortment selection in settings with fixed prices, we consider the price as a hyper-parameter and run the algorithm with the best selection of fixed pricing. On the other hand, M3P is designed to optimize prices under the assumption that all $n$ items can be offered without any need for assortment selection. To account for the assortment size limitations of our experimental setting, we consider a version of this algorithm that only offers top $K$ items (based on their estimated utility value) under the prices chosen by M3P. Figure 2 illustrates that our algorithms, which simultaneously address both assortment selection and pricing, outperform methods that concentrate solely on either assortment selection or pricing.

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

## A  PROOF OF PROPOSITION 2

For each item $i \in S_t \cup \{0\}$, we define the choice response variables $y_{ti} = \mathbf{1}\{i_t = i\} \in \{0, 1\}$. Furthermore, let us denote $q_{ti}(\boldsymbol{\theta}) = q_t(i|S_t, \boldsymbol{p}_t; \boldsymbol{\theta})$ as the $S_t$ and $\boldsymbol{p}_t$ dependency is clear from the context throughout the proof. Then, the gradient of these probabilities with respect to $\boldsymbol{\theta}$ can be written as

$$\nabla_{\boldsymbol{\theta}} q_{ti}(\boldsymbol{\theta}) = q_{ti}(\boldsymbol{\theta}) \left( \widetilde{\boldsymbol{x}}_{ti} - \sum_{j \in S_t} q_{tj}(\boldsymbol{\theta}) \widetilde{\boldsymbol{x}}_{tj} \right).$$

On the other hand, we can write the negative log-likelihood function at time $t$ as

$$\ell_t(\boldsymbol{\theta}) := -\sum_{\tau=1}^{t-1} \sum_{i \in S_\tau \cup \{0\}} y_{ti} \log q_{\tau i}(\boldsymbol{\theta}).$$

Calculating the gradient of this negative log-likelihood with respect to $\boldsymbol{\theta}$ we obtain

$$\nabla_{\boldsymbol{\theta}} \ell_t(\boldsymbol{\theta}) = \sum_{\tau=1}^{t-1} \sum_{i \in S_\tau} (q_{\tau i}(\boldsymbol{\theta}) - y_{\tau i}) \widetilde{\boldsymbol{x}}_{\tau i}$$

On the other hand, the Hessian of the negative log-likelihood is given by

$$\nabla_{\boldsymbol{\theta}}^2 \ell_t(\boldsymbol{\theta}) = \sum_{\tau=1}^{t-1} \sum_{i \in S_\tau} q_{\tau i}(\boldsymbol{\theta}) \widetilde{\boldsymbol{x}}_{\tau i} \left( \widetilde{\boldsymbol{x}}_{\tau i} - \sum_{j \in S_t} q_{\tau j}(\boldsymbol{\theta}) \widetilde{\boldsymbol{x}}_{\tau j} \right)^\top$$

$$= \sum_{\tau=1}^{t-1} \left( \sum_{i \in S_\tau} q_{ti}(\boldsymbol{\theta}) \widetilde{\boldsymbol{x}}_{\tau i} \widetilde{\boldsymbol{x}}_{\tau i}^\top - \sum_{i \in S_\tau} \sum_{j \in S_t} q_{ti}(\boldsymbol{\theta}) q_{\tau j}(\boldsymbol{\theta}) \widetilde{\boldsymbol{x}}_{\tau i} \widetilde{\boldsymbol{x}}_{\tau j}^\top \right).$$

Now, let $\boldsymbol{q}_t(\boldsymbol{\theta})$ denote the vector of probabilities $q_{ti}(\boldsymbol{\theta})$ and let $\widetilde{\boldsymbol{X}}_t$ be the matrix with columns $\widetilde{\boldsymbol{x}}_{ti}$ for $i \in S_t$, we can write

$$\nabla_{\boldsymbol{\theta}}^2 \ell_t(\boldsymbol{\theta}) = \sum_{\tau=1}^{t-1} \widetilde{\boldsymbol{X}}_\tau \Sigma_\tau(\boldsymbol{\theta}) \widetilde{\boldsymbol{X}}_\tau^\top.$$

where $\Sigma_t(\boldsymbol{\theta}) = \operatorname{diag}(\boldsymbol{q}_t(\boldsymbol{\theta})) - \boldsymbol{q}_t(\boldsymbol{\theta}) \boldsymbol{q}_t(\boldsymbol{\theta})^\top$. Since we have $q_{ti}(\boldsymbol{\theta}) q_{t0}(\boldsymbol{\theta}) > 0$ for all $\boldsymbol{\theta} \in \mathbb{R}^{2d}$, we conclude that $\Sigma_t(\boldsymbol{\theta})) \succ \mathbf{0}$ for all $\boldsymbol{\theta} \in \mathbb{R}^{2d}$.

Therefore, $\nabla_{\boldsymbol{\theta}}^2 \ell_t(\boldsymbol{\theta}) \succcurlyeq \mathbf{0}$ for all $\boldsymbol{\theta} \in \mathbb{R}^{2d}$. Hence, the negative log-likelihood is convex with respect to $\boldsymbol{\theta}$. As a result, any $\boldsymbol{\theta}$ that satisfies the first-order optimality condition $\nabla_{\boldsymbol{\theta}} \ell_t(\boldsymbol{\theta}) = 0$ is a minimizer.

Furthermore, if we are given that feature covariance matrix $V_{t-1} = \sum_{\tau=1}^{t-1} \sum_{i \in S_\tau} \widetilde{\boldsymbol{x}}_{ti} \widetilde{\boldsymbol{x}}_{ti}^\top$ is positive definite, i.e. $V_{t-1} \succ \mathbf{0}$, the negative log-likelihood function becomes strongly convex with respect to $\boldsymbol{\theta}$. Consequently, we have a unique MLE solution.

# B PROOF OF THEOREM 3

We start by recalling Proposition 1 which defines $B_t$ as the unique solution of

$$B = \max_{S \in \mathcal{S}_K} \sum_{i \in S} v_{ti}(B). \tag{9}$$

and asserts that the optimum prices are $p_{ti}^* = 1/\beta_{ti} + B_t$. Our first lemma shows that this fixed point $B_t$ lies within $[0, P_0]$ for some $P_0$ under our assumptions, allowing us to constrain our search for the fixed point into a bounded interval. This result also implies that the optimum prices $p_{ti}^*$ are bounded within $[1, P]$ for some $P$.

**Lemma 1** (Bounded optimum prices). *Consider that the utility function for each item $i \in [n]$ is given by $u_{ti}(p) = \alpha_{ti} - \beta_{ti} \cdot p$ for some $\alpha_{ti} \leq 1$ and $L_0 \leq \beta_{ti} \leq 1$. Then, $B_t \in [0, P_0]$ and $p_{ti}^* \in [1, P]$ for constants $P_0$ and $P$ that only depend on $K$ and $L_0$.*

Recall that $T_0$ is the length of random initialization. At each round $t \leq T_0$, the algorithm chooses a subset $S_t$ uniformly at random from $\{S \subseteq [n] : |S| \leq K\}$ and sets $p_{ti} \in [1, 2]$ uniformly at random for all $i \in S_t$. Then, using the assumption that there exists a constant $\sigma_0 > 0$ such that $\mathbb{E}[\boldsymbol{x}_{ti} \boldsymbol{x}_{ti}^\top] \succeq \sigma_0 \boldsymbol{I}$, we can show the following lemma regarding initialization.

**Lemma 2** (Initialization). *Define $V_{T_0} = \sum_{t=1}^{T_0} \sum_{i \in S_t} \widetilde{\boldsymbol{x}}_{ti} \widetilde{\boldsymbol{x}}_{ti}^\top$. There exist some positive, universal constants $C_1$ and $C_2$ such that if the length of random initialization satisfies*

$$T_0 \geq \frac{1}{K} \left( \frac{C_1 \sqrt{d} + C_2 \sqrt{\log T}}{\sigma_0} \right)^2 + \frac{2B}{K \sigma_0},$$

*then $\lambda_{\min}(V_{T_0}) \geq B$ with probability at least $1 - \frac{1}{T^2}$.*

Lemma 2 implies that we can have $\lambda_{\min}(V_{T_0}) \geq K(1 + P^2)$ with a high probability if we run the random initialization for $\mathcal{O}(\sigma_0^{-2}(d + \log T + P^2))$ rounds. Similar to Li et al. (2017) and Oh & Iyengar (2021), the independence assumption (Assumption 3) on the feature vectors $\boldsymbol{x}_{ti}$ is only needed to ensure that $V_{T_0}$ is invertible at the end of the initialization phase. We do not require this stochasticity assumption in the rest of the regret analysis. Therefore, after the random initialization period of the first $T_0$ rounds, the context vectors $\boldsymbol{x}_{ti}$ can even be chosen adversarially as long as their norms $\|\boldsymbol{x}_{ti}\|$ are bounded and they satisfy the minimum price sensitivity condition $\langle \boldsymbol{\phi}^*, \boldsymbol{x}_{ti} \rangle \geq L_0$.

The following lemma shows that the assortments $S_t$ and prices $\boldsymbol{p}_t$ offered by Algorithm 2 at any round $t$ results in a sufficiently large probability of selection for any item $i \in S_t$ under any parameter $\boldsymbol{\theta}$ sufficiently close to $\boldsymbol{\theta}^*$. This condition is central in showing that the maximum likelihood estimator is consistent (Lemma 4) and satisfies a finite-sample normality-type estimation error bound (Lemma 5).

**Lemma 3.** *Let $\boldsymbol{\theta}$ be some parameter such that $\|\boldsymbol{\theta} - \boldsymbol{\theta}^*\| \leq 1$. Then, there exists some constant $\kappa > 0$ that depends only on $K$ and $L_0$ such that*

$$q_t(i|S_t, \boldsymbol{p}_t; \boldsymbol{\theta}) q_t(0|S_t, \boldsymbol{p}_t; \boldsymbol{\theta}) \geq \kappa$$

*for all $i \in S_t$ at any round $t \in [T]$.*

**Lemma 4** (Consistency of MLE). *Let $T_0$ be any round such that*

$$\lambda_{\min}(V_{T_0}) \geq \max \left\{ \frac{16}{\kappa^2}(4d + 2\log(T)), K(1 + P^2) \right\}.$$

*Then for any $t \geq T_0$, $\mathbb{P}(\|\widehat{\boldsymbol{\theta}}_t - \boldsymbol{\theta}^*\| > 1) \leq 1 - \frac{1}{T^2}$.*

Combining the results of Lemma 2 and Lemma 4, we can show that the condition $\|\widehat{\boldsymbol{\theta}}_t - \boldsymbol{\theta}^*\| \leq 1$ is satisfied for any $t \geq T_0$ with probability $1 - \mathcal{O}(T^{-2})$ if we select

$$T_0 = \Theta \left( \frac{d + \log T + P^2}{\sigma_0^2 \kappa^2} \right). \tag{10}$$

Thus, we can define a *good* event

$$\mathcal{E}_0 = \left\{ \|\widehat{\boldsymbol{\theta}}_t - \boldsymbol{\theta}^*\| \leq 1, \forall t \geq T_0 \right\}$$

and show that it holds with probability $1 - \mathcal{O}(T^{-1})$.

**Lemma 5** (Normality of MLE). *Suppose* $\|\widehat{\boldsymbol{\theta}}_t - \boldsymbol{\theta}^*\| \leq 1$ *for* $t \geq T_0$. *Then*

$$\|\widehat{\boldsymbol{\theta}}_t - \boldsymbol{\theta}^*\|_{V_t} \leq \frac{2}{\kappa} \sqrt{d \log\left(\frac{t}{2d}\right) + 2\log t} \tag{11}$$

*with probability at least* $1 - t^{-2}$.

For the selection of $T_0$ given in equation 10, we already showed that $\mathcal{E}_0$ holds with probability $1 - \mathcal{O}(T^{-1})$. Therefore, conditioned on $\mathcal{E}_0$ happens, we can further ensure that $\|\widehat{\boldsymbol{\theta}}_t - \boldsymbol{\theta}^*\|_{V_t} \leq \alpha_t$ holds with probability at least $1 - t^{-2}$ if we choose the confidence radius as

$$\alpha_t = \frac{2}{\kappa} \sqrt{d \log\left(\frac{t}{2d}\right) + 2\log t}. \tag{12}$$

In other words, for any $t \geq T_0$, we can define another *good* event $\mathcal{E}_t = \{\|\widehat{\boldsymbol{\theta}}_t - \boldsymbol{\theta}^*\|_{V_t} \leq \alpha_t\}$ that holds with probability at least $1 - t^{-2}$ conditioned on $\mathcal{E}_0$.

Next, the following lemma establishes important properties for the optimistic utility functions constructed in Algorithm 2.

**Lemma 6.** *Assume that* $\|\widehat{\boldsymbol{\theta}}_t - \boldsymbol{\theta}^*\|_{V_t} \leq \alpha_t$ *holds for some* $\alpha_t$ *and let* $h_{ti} : \mathbb{R} \to \mathbb{R}$ *be the linear function defined as*

$$h_{ti}(p) = \min\{\langle \widehat{\boldsymbol{\psi}}_t, \boldsymbol{x}_{ti} \rangle + g_{ti}, 1\} - \max\{\langle \widehat{\boldsymbol{\phi}}_t, \boldsymbol{x}_{ti} \rangle - g_{ti}, L_0\} \cdot p,$$

*where* $\widehat{\boldsymbol{\theta}}_t = (\widehat{\boldsymbol{\psi}}_t, \widehat{\boldsymbol{\phi}}_t)$ *and* $g_{ti} := \alpha_t \|(\boldsymbol{x}_{ti}, \boldsymbol{x}_{ti})\|_{V_t^{-1}}$. *Then,* $h_{ti}(p)$ *satisfies*

$$h_{ti}(p) \geq u_{ti}(p), \quad \forall p \geq 0 \tag{13}$$
$$h_{ti}(p) - u_{ti}(p) \leq 3Pg_{ti}, \quad \forall p \in [1, P]. \tag{14}$$

We note that Algorithm 2 chooses the assortment $S_t$ and prices $\boldsymbol{p}_t$ by solving

$$(S_t, \boldsymbol{p}_t) \in \underset{\substack{S \in \mathcal{S}_K \\ \boldsymbol{p} \in \mathbb{R}_+^n}}{\arg\max} \widetilde{R}_t(S, \boldsymbol{p})$$

where $\widetilde{R}_t(S, \boldsymbol{p})$ denotes the optimistic estimate of the revenue function as defined in equation 7. Then, using the properties of the optimistic estimate of the utility functions $h_{ti}(p)$, we can show the following lemma.

**Lemma 7.** *Assume good event* $\mathcal{E}$ *occurs. Then, for any* $t \in [T]$, *it holds that*

(a) $R_t(S_t^*, \boldsymbol{p}_t^*) \leq \widetilde{R}_t(S_t, \boldsymbol{p}_t)$, *and*

(b) $\widetilde{R}_t(S_t, \boldsymbol{p}_t) - R_t(S_t, \boldsymbol{p}_t) \leq 3P^2 \max_{i \in S_t} g_{ti}$.

Now, we break the regret $\mathcal{R}_T$ into the initialization phase and the learning phase:

$$\mathcal{R}_T = \mathbb{E}\left[\sum_{t=1}^{T_0} (R_t(S_t^*, \boldsymbol{p}_t^*) - R_t(S_t, \boldsymbol{p}_t))\right] + \mathbb{E}\left[\sum_{t=T_0+1}^{T} (R_t(S_t^*, \boldsymbol{p}_t^*) - R_t(S_t, \boldsymbol{p}_t))\right]$$

$$\leq PT_0 + \mathbb{E}\left[\sum_{t=T_0+1}^{T} \left(\widetilde{R}_t(S_t, \boldsymbol{p}_t) - R_t(S_t, \boldsymbol{p}_t)\right)\right]$$

where the last inequality follows from property (a) in Lemma 7. Now, we decompose the expectation term into two parts where the high probability event $\mathcal{E}_0$ holds and it does not.

$$\mathcal{R}_T \leq PT_0 + \mathbb{E}\left[\sum_{t=T_0+1}^{T} \left(\widetilde{R}_t(S_t, \boldsymbol{p}_t) - R_t(S_t, \boldsymbol{p}_t)\right) \mathbf{1}(\mathcal{E}_0)\right] + \mathbb{E}\left[\sum_{t=T_0+1}^{T} \left(\widetilde{R}_t(S_t, \boldsymbol{p}_t) - R_t(S_t, \boldsymbol{p}_t)\right) \mathbf{1}(\neg\mathcal{E}_0)\right]$$

$$\leq PT_0 + \sum_{t=T_0+1}^{T} \mathbb{E}\left[\left(\widetilde{R}_t(S_t, \boldsymbol{p}_t) - R_t(S_t, \boldsymbol{p}_t)\right) \mathbf{1}(\mathcal{E}_0)\right] + \mathcal{O}(P).$$

For each expectation term in the remaining summation, we can split it into two parts where the high probability event $\mathcal{E}_t$ holds and it does not:

$$\mathbb{E}\left[\left(\widetilde{R}_t(S_t, \boldsymbol{p}_t) - R_t(S_t, \boldsymbol{p}_t)\right) \mathbf{1}(\mathcal{E}_0)\right]$$
$$= \mathbb{E}\left[\left(\widetilde{R}_t(S_t, \boldsymbol{p}_t) - R_t(S_t, \boldsymbol{p}_t)\right) \mathbf{1}(\mathcal{E}_0)\mathbf{1}(\mathcal{E}_t)\right] + \mathbb{E}\left[\left(\widetilde{R}_t(S_t, \boldsymbol{p}_t) - R_t(S_t, \boldsymbol{p}_t)\right) \mathbf{1}(\mathcal{E}_0)\mathbf{1}(\neg\mathcal{E}_t)\right]$$
$$\leq 3P^2\alpha_t \max_{i\in S_t}\|(\boldsymbol{x}_{ti}, \boldsymbol{x}_{ti})\|_{V_t^{-1}} + \mathcal{O}(P \cdot t^{-2}).$$

where the last inequality follows from property (b) in Lemma 7. As a result,

$$\mathcal{R}_T \leq PT_0 + 3P^2\sum_{t=1}^{T} \alpha_t \max_{i\in S_t}\|(\boldsymbol{x}_{ti}, \boldsymbol{x}_{ti})\|_{V_t^{-1}} + \sum_{t=1}^{T}\mathcal{O}(P \cdot t^{-2}) + \mathcal{O}(P)$$

$$\leq PT_0 + 3P^2\sum_{t=1}^{T} \alpha_t \max_{i\in S_t}\|(\boldsymbol{x}_{ti}, \boldsymbol{x}_{ti})\|_{V_t^{-1}} + \mathcal{O}(P)$$

Applying Cauchy-Schwarz inequality in the second term, it follows that

$$\mathcal{R}_T \leq PT_0 + 3P^2\alpha_T\sqrt{T\sum_{t=1}^{T} \max_{i\in S_t}\|(\boldsymbol{x}_{ti}, \boldsymbol{x}_{ti})\|_{V_t^{-1}}} + \mathcal{O}(P).$$

Applying Lemma 10 and plugging in the value for $\alpha_T$, we obtain

$$\mathcal{R}_T \leq PT_0 + \frac{12P^2}{\kappa}\sqrt{Td\log(T/d)\left(d\log\left(\frac{T}{2d}\right) + 2\log T\right)} + \mathcal{O}(P).$$

### B.1    PROOFS FOR TECHNICAL LEMMAS

**Lemma 1** (Bounded optimum prices). *Consider that the utility function for each item $i \in [n]$ is given by $u_{ti}(p) = \alpha_{ti} - \beta_{ti} \cdot p$ for some $\alpha_{ti} \leq 1$ and $L_0 \leq \beta_{ti} \leq 1$. Then, $B_t \in [0, P_0]$ and $p_{ti}^* \in [1, P]$ for constants $P_0$ and $P$ that only depend on $K$ and $L_0$.*

*Proof.* Let $B_u$ be the unique solution of the fixed point equation

$$B = \frac{K}{L_0}e^{-L_0 B}. \tag{15}$$

Given the bounds on $\alpha_{ti}$ and $\beta_{ti}$, we notice that the right-hand side of equation 15 is an upper bound for the right-hand side of equation 5, hence $B_t \leq B_u$.

In equation 15, the left-hand side is increasing and the right-hand side is decreasing in $B$. Additionally, for $B = 1 + 2\max\{1, \log(K)\}/L_0$, the left-hand side of equation 15 is greater than the right-hand side. Hence, the fixed point satisfies $0 \leq B_t \leq B_u \leq P_0 = 1 + 2\max\{1, \log(K)\}/L_0$.

Furthermore, we have $1 \leq 1/\beta_{ti} \leq 1/L_0$. Therefore, $1 \leq p_{ti}^* \leq 1 + (1 + 2\max\{1, \log(K)\})/L_0$.

$\square$

**Lemma 2** (Initialization). *Define $V_{T_0} = \sum_{t=1}^{T_0} \sum_{i \in S_t} \widetilde{\boldsymbol{x}}_{ti} \widetilde{\boldsymbol{x}}_{ti}^\top$. There exist some positive, universal constants $C_1$ and $C_2$ such that if the length of random initialization satisfies*

$$T_0 \geq \frac{1}{K} \left( \frac{C_1\sqrt{d} + C_2\sqrt{\log T}}{\sigma_0} \right)^2 + \frac{2B}{K\sigma_0},$$

*then $\lambda_{\min}(V_{T_0}) \geq B$ with probability at least $1 - \frac{1}{T^2}$.*

*Proof.* Let $\boldsymbol{\Sigma} = \mathbb{E}[\boldsymbol{x}_{ti}\boldsymbol{x}_{ti}^\top]$ and $\widetilde{\boldsymbol{\Sigma}} = \mathbb{E}[\widetilde{\boldsymbol{x}}_{ti}\widetilde{\boldsymbol{x}}_{ti}^\top]$. Then, noting that $p_{ti}$ is uniformly distributed, we have

$$\widetilde{\boldsymbol{\Sigma}} = \begin{bmatrix} \boldsymbol{\Sigma} & -\frac{3}{2}\boldsymbol{\Sigma} \\ -\frac{3}{2}\boldsymbol{\Sigma} & \frac{7}{3}\boldsymbol{\Sigma} \end{bmatrix}.$$

Then, using Schur's formula, each eigenvalue $\widetilde{\lambda}$ of $\widetilde{\boldsymbol{\Sigma}}$ are given by solutions of the equation

$$0 = \det(\widetilde{\boldsymbol{\Sigma}} - \widetilde{\lambda}\boldsymbol{I})$$
$$= \det(\boldsymbol{\Sigma} - \widetilde{\lambda}\boldsymbol{I}) \det\left( \frac{7}{3}\boldsymbol{\Sigma} - \widetilde{\lambda}\boldsymbol{I} - \frac{9}{4}\boldsymbol{\Sigma}(\boldsymbol{\Sigma} - \widetilde{\lambda}\boldsymbol{I})^{-1}\boldsymbol{\Sigma} \right).$$

Since the inverse of the matrix $\boldsymbol{\Sigma} - \widetilde{\lambda}\boldsymbol{I}$ appears on the right-hand side, we must have $\det(\boldsymbol{\Sigma} - \widetilde{\lambda}\boldsymbol{I}) \neq 0$. Hence, all eigenvalues must satisfy

$$\det\left( \frac{7}{3}\boldsymbol{\Sigma} - \widetilde{\lambda}\boldsymbol{I} - \frac{9}{4}\boldsymbol{\Sigma}(\boldsymbol{\Sigma} - \widetilde{\lambda}\boldsymbol{I})^{-1}\boldsymbol{\Sigma} \right) = 0.$$

Letting $\boldsymbol{\Sigma} = \boldsymbol{V}\boldsymbol{\Lambda}\boldsymbol{V}^\top$ be the eigen-decomposition of $\boldsymbol{\Sigma}$ with $\{\lambda_j\}_{j=1}^d$ denoting the eigenvalues. Then, we can write

$$0 = \det\left( \frac{7}{3}\boldsymbol{V}\boldsymbol{\Lambda}\boldsymbol{V}^\top - \widetilde{\lambda}\boldsymbol{I} - \frac{9}{4}\boldsymbol{V}\boldsymbol{\Lambda}\boldsymbol{V}^\top(\boldsymbol{V}\boldsymbol{\Lambda}\boldsymbol{V}^\top - \widetilde{\lambda}\boldsymbol{I})^{-1}\boldsymbol{V}\boldsymbol{\Lambda}\boldsymbol{V}^\top \right)$$
$$= \det(\boldsymbol{V})^2 \det\left( \frac{7}{3}\boldsymbol{\Lambda} - \widetilde{\lambda}\boldsymbol{I} - \frac{9}{4}\boldsymbol{\Lambda}(\boldsymbol{\Lambda} - \widetilde{\lambda}\boldsymbol{I})^{-1}\boldsymbol{\Lambda} \right)$$
$$= \prod_{j=1}^d \left( \frac{7}{3}\lambda_j - \widetilde{\lambda} - \frac{9}{4}\frac{\lambda_j^2}{\lambda_j - \widetilde{\lambda}} \right).$$

Consequently, the eigenvalues of $\widetilde{\boldsymbol{\Sigma}}$ are given by

$$\widetilde{\lambda}_{j,1} = (20 + 2\sqrt{97})\lambda_j \text{ and } \widetilde{\lambda}_{j,2} = (20 - 2\sqrt{97})\lambda_j, \ \forall j \in [d].$$

Since $\lambda_j \geq \sigma_0$ for all $j$ by Assumption 3, $\lambda_{\min}(\widetilde{\boldsymbol{\Sigma}}) \geq C\sigma_0$ for some positive, universal constant $C$. $\qquad\square$

**Lemma 3.** *Let $\boldsymbol{\theta}$ be some parameter such that $\|\boldsymbol{\theta} - \boldsymbol{\theta}^*\| \leq 1$. Then, there exists some constant $\kappa > 0$ that depends only on $K$ and $L_0$ such that*

$$q_t(i|S_t, \boldsymbol{p}_t; \boldsymbol{\theta})q_t(0|S_t, \boldsymbol{p}_t; \boldsymbol{\theta}) \geq \kappa$$

*for all $i \in S_t$ at any round $t \in [T]$.*

*Proof.* Note that for all $t \leq T_0$, we have $1 \leq p_{ti} \leq 2$. Additionally, by Lemma 1, the prices at all rounds $t > T_0$ satisfy $1 \leq p_{ti} \leq P$ for some constant $P$ that depends only on $L_0$ and $K$. Therefore, we can conclude that $1 \leq p_{ti} \leq \max\{2, P\}$ for all $t \in [T]$ and $i \in S_t$.

Next, we let $\boldsymbol{\theta} = (\boldsymbol{\psi}, \boldsymbol{\phi})$ and $\boldsymbol{\epsilon} = \boldsymbol{\theta} - \boldsymbol{\theta}^*$ with $\|\boldsymbol{\epsilon}\| \leq 1$. As a result, we can show that

$$-2 \leq \langle \boldsymbol{\psi}, \boldsymbol{x}_{ti} \rangle \leq 2 \quad \text{and}$$
$$-1 \leq \langle \boldsymbol{\phi}, \boldsymbol{x}_{ti} \rangle \leq 2.$$

As a result, we can write

$$q_t(i|S_t, \boldsymbol{p}_t; \boldsymbol{\theta})q_t(0|S_t, \boldsymbol{p}_t; \boldsymbol{\theta}) = \frac{\exp(\langle \boldsymbol{\psi}, \boldsymbol{x}_{ti} \rangle - \langle \boldsymbol{\phi}, \boldsymbol{x}_{ti} \rangle p_{ti})}{\left(1 + \sum_{j \in S_t} \exp(\langle \boldsymbol{\psi}, \boldsymbol{x}_{tj} \rangle - \langle \boldsymbol{\phi}, \boldsymbol{x}_{tj} \rangle p_{tj})\right)^2}$$
$$\geq \frac{e^{-2(1+\max\{2, P\})}}{(1 + Ke^3)^2}$$

to show that $q_t(i|S_t, \boldsymbol{p}_t; \boldsymbol{\theta})q_t(0|S_t, \boldsymbol{p}_t; \boldsymbol{\theta})$ is lower bounded by some constant $\kappa > 0$ that depends only on $L_0$ and $K$. $\qquad\square$

**Lemma 4** (Consistency of MLE). *Let $T_0$ be any round such that*

$$\lambda_{\min}(V_{T_0}) \geq \max \left\{ \frac{16}{\kappa^2}(4d + 2\log(T)), K(1 + P^2) \right\}.$$

*Then for any $t \geq T_0$, $\mathbb{P}(\|\widehat{\boldsymbol{\theta}}_t - \boldsymbol{\theta}^*\| > 1) \leq 1 - \frac{1}{T^2}$.*

*Proof.* Calculating the gradient of this negative log-likelihood with respect to $\boldsymbol{\theta}$ we obtain

$$\nabla_{\boldsymbol{\theta}} \ell_t(\boldsymbol{\theta}) = \sum_{\tau=1}^{t-1} \sum_{i \in S_\tau} (q_{\tau i}(\boldsymbol{\theta}) - y_{\tau i}) \widetilde{\boldsymbol{x}}_{\tau i}.$$

Then, let us denote the expectation of $\nabla_{\boldsymbol{\theta}} \ell_t(\boldsymbol{\theta})$ over the user choices $i_t$ by

$$G_t(\boldsymbol{\theta}) := \sum_{\tau=1}^{t-1} \sum_{i \in S_\tau} (q_\tau(i|S_\tau, \boldsymbol{p}_\tau; \boldsymbol{\theta}) - q_\tau(i|S_\tau, \boldsymbol{p}_\tau; \boldsymbol{\theta}^*)) \widetilde{\boldsymbol{x}}_{\tau i},$$

and we have

$$G_t(\boldsymbol{\theta}^*) = 0 \text{ and } G_t(\widehat{\boldsymbol{\theta}}_n) = \sum_{\tau=1}^{t-1} \sum_{i \in S_\tau} \epsilon_{\tau i} \widetilde{\boldsymbol{x}}_{\tau i}.$$

where $\epsilon_{ti} = y_{ti} - q_t(i|S_t, \boldsymbol{p}_t; \boldsymbol{\theta}^*)$ are sub-Gaussian random variables with parameter 1. Note that collections of variables $\{y_{ti}\}_{i \in S_t}$ are independent over $t$, but the variables within each collection are not independent.

For brevity, we will denote $q_{ti}(\boldsymbol{\theta}) := q_t(i|S_t, \boldsymbol{p}_t; \boldsymbol{\theta})$ when it is clear that $S_t$ and $\boldsymbol{p}_t$ are selected assortment and prices at time $t$.

For any $\boldsymbol{\theta}_1, \boldsymbol{\theta}_2 \in \mathbb{R}^d$, mean value theorem implies that there exists some $\bar{\boldsymbol{\theta}} = \lambda \boldsymbol{\theta}_1 + (1 - \lambda)\boldsymbol{\theta}_2$ with $0 < \lambda < 1$, such that

$$G(\boldsymbol{\theta}_1) - G(\boldsymbol{\theta}_2) = F(\bar{\boldsymbol{\theta}})(\boldsymbol{\theta}_1 - \boldsymbol{\theta}_2)$$

where we defined

$$F(\bar{\boldsymbol{\theta}}) := \left[ \sum_{\tau=1}^{t-1} \sum_{i \in S_\tau} \widetilde{\boldsymbol{x}}_{\tau i} \nabla_{\boldsymbol{\theta}} q_{\tau i}(\bar{\boldsymbol{\theta}}) \right].$$

Let $H_\tau := \sum_{i \in S_\tau} \widetilde{\boldsymbol{x}}_{\tau i} \nabla_{\boldsymbol{\theta}} q_{\tau i}(\overline{\boldsymbol{\theta}})$ and notice that

$$
\begin{aligned}
H_\tau &= \sum_{i \in S_\tau} q_{\tau i}(\overline{\boldsymbol{\theta}}) \widetilde{\boldsymbol{x}}_{\tau i} \widetilde{\boldsymbol{x}}_{\tau i}^\top - \sum_{i \in S_\tau} \sum_{j \in S_\tau} q_{\tau i}(\overline{\boldsymbol{\theta}}) q_{\tau j}(\overline{\boldsymbol{\theta}}) \widetilde{\boldsymbol{x}}_{\tau i} \widetilde{\boldsymbol{x}}_{\tau j}^\top \\
&= \sum_{i \in S_\tau} q_{\tau i}(\overline{\boldsymbol{\theta}}) \widetilde{\boldsymbol{x}}_{\tau i} \widetilde{\boldsymbol{x}}_{\tau i}^\top - \frac{1}{2} \sum_{i \in S_\tau} \sum_{j \in S_\tau} q_{\tau i}(\overline{\boldsymbol{\theta}}) q_{\tau j}(\overline{\boldsymbol{\theta}}) (\widetilde{\boldsymbol{x}}_{\tau i} \widetilde{\boldsymbol{x}}_{\tau j}^\top + \widetilde{\boldsymbol{x}}_{\tau j} \widetilde{\boldsymbol{x}}_{\tau i}^\top) \\
&\succcurlyeq \sum_{i \in S_\tau} q_{\tau i}(\overline{\boldsymbol{\theta}}) \widetilde{\boldsymbol{x}}_{\tau i} \widetilde{\boldsymbol{x}}_{\tau i}^\top - \frac{1}{2} \sum_{i \in S_\tau} \sum_{j \in S_\tau} q_{\tau i}(\overline{\boldsymbol{\theta}}) q_{\tau j}(\overline{\boldsymbol{\theta}}) (\widetilde{\boldsymbol{x}}_{\tau i} \widetilde{\boldsymbol{x}}_{\tau i}^\top + \widetilde{\boldsymbol{x}}_{\tau j} \widetilde{\boldsymbol{x}}_{\tau j}^\top) \\
&= \sum_{i \in S_\tau} q_{\tau i}(\overline{\boldsymbol{\theta}}) \widetilde{\boldsymbol{x}}_{\tau i} \widetilde{\boldsymbol{x}}_{\tau i}^\top - \sum_{i \in S_\tau} \sum_{j \in S_\tau} q_{\tau i}(\overline{\boldsymbol{\theta}}) q_{\tau j}(\overline{\boldsymbol{\theta}}) \widetilde{\boldsymbol{x}}_{\tau i} \widetilde{\boldsymbol{x}}_{\tau i}^\top \\
&= \sum_{i \in S_\tau} q_{\tau i}(\overline{\boldsymbol{\theta}}) \left( 1 - \sum_{j \in S_\tau} q_{\tau j}(\overline{\boldsymbol{\theta}}) \right) \widetilde{\boldsymbol{x}}_{\tau i} \widetilde{\boldsymbol{x}}_{\tau i}^\top \\
&= \sum_{i \in S_\tau} q_{\tau i}(\overline{\boldsymbol{\theta}}) q_{\tau 0}(\overline{\boldsymbol{\theta}}) \widetilde{\boldsymbol{x}}_{\tau i} \widetilde{\boldsymbol{x}}_{\tau i}^\top.
\end{aligned}
$$

Define $\mathcal{H}_t(\overline{\boldsymbol{\theta}}) = \sum_{\tau=1}^{t-1} \sum_{i \in S_\tau} q_{\tau i}(\overline{\boldsymbol{\theta}}) q_{\tau 0}(\overline{\boldsymbol{\theta}}) \widetilde{\boldsymbol{x}}_{\tau i} \widetilde{\boldsymbol{x}}_{\tau i}^\top$ and observe that

$$
G(\boldsymbol{\theta}_1) - G(\boldsymbol{\theta}_2) = F(\overline{\boldsymbol{\theta}})(\boldsymbol{\theta}_1 - \boldsymbol{\theta}_2) \geq \mathcal{H}_t(\overline{\boldsymbol{\theta}})(\boldsymbol{\theta}_1 - \boldsymbol{\theta}_2).
$$

Now, define $\mathcal{B}_\eta := \{\boldsymbol{\theta} : \|\boldsymbol{\theta} - \boldsymbol{\theta}^*\| \leq \eta\}$ and $\kappa_\eta := \inf_{\boldsymbol{\theta} \in \mathcal{B}_\eta} q_{\tau i}(\boldsymbol{\theta}) q_{\tau 0}(\boldsymbol{\theta}) > 0$. Since we have $\mathcal{H}_t(\boldsymbol{\theta}) \succcurlyeq \kappa_\eta V_t$ for any $\theta \in \mathcal{B}_\eta$, and $G(\boldsymbol{\theta})$ is an injection from $\mathbb{R}^{2d}$ to $\mathbb{R}^{2d}$; Lemma A of Chen et al. (1999) implies that

$$
\left\{ \boldsymbol{\theta} : \|G(\boldsymbol{\theta})\|_{V_t^{-1}} \leq \kappa_\eta \eta \sqrt{\lambda_{\min}(V_t)} \right\} \subseteq \mathcal{B}_\eta.
$$

In addition, Lemma 15 of Oh & Iyengar (2021) shows that

$$
\mathcal{E}_G := \|G(\widehat{\boldsymbol{\theta}}_n)\|_{V_t^{-1}} \leq 2\sqrt{4d + \log(1/\delta)}
$$

holds with probability at least $1 - \delta$. Suppose $\mathcal{E}_G$ holds for the rest of the proof. Then, $\lambda_{\min}(V_t) \geq \frac{16}{\kappa^2 \eta^2}(4d + \log(1/\delta))$ implies that $\widehat{\boldsymbol{\theta}}_n \in \mathcal{B}_\eta$.

Since $\kappa = \kappa_1$, we have $\kappa_\eta \geq \kappa$ for all $\eta \leq 1$. Thus, we have $\|\widehat{\boldsymbol{\theta}}_n - \boldsymbol{\theta}^*\| \leq 1$ when $\lambda_{\min}(V_t) \geq \frac{16}{\kappa^2}(4d + \log(1/\delta))$.

$\square$

**Lemma 5** (Normality of MLE). *Suppose* $\|\widehat{\boldsymbol{\theta}}_t - \boldsymbol{\theta}^*\| \leq 1$ *for* $t \geq T_0$*. Then*

$$
\|\widehat{\boldsymbol{\theta}}_t - \boldsymbol{\theta}^*\|_{V_t} \leq \frac{2}{\kappa} \sqrt{d \log\left(\frac{t}{2d}\right) + 2 \log t} \tag{11}
$$

*with probability at least* $1 - t^{-2}$*.*

*Proof.* Following the proof of Lemma 4, we obtain

$$
\|G(\widehat{\boldsymbol{\theta}}_t)\|_{V_t^{-1}}^2 \geq \kappa^2 \|\widehat{\boldsymbol{\theta}}_t - \boldsymbol{\theta}^*\|_{V_t}^2
$$

for any $\widehat{\boldsymbol{\theta}}_t \in \{\boldsymbol{\theta} : \|\boldsymbol{\theta} - \boldsymbol{\theta}^*\| \leq 1\}$. Then, we use Theorem 1 in Abbasi-Yadkori (2011), which states if the noise $\epsilon_{ti}$ is sub-Gaussian with parameter 1, then

$$
\|G(\widehat{\boldsymbol{\theta}}_t)\|_{V_t^{-1}}^2 \leq 2 \log\left(\frac{\det(V_t)^{1/2} \det(V_{T_0})^{-1/2}}{\delta}\right)
$$

with probability at least $1 - \delta$. Then we combine with Lemma 8 to obtain

$$\|G(\widehat{\boldsymbol{\theta}}_t)\|^2_{V_t^{-1}} \leq 2\left[d \log\left(\frac{tK(1 + P^2)}{2d}\right) - \frac{1}{2}\log\det(V_{T_0}) + \log\frac{1}{\delta}\right]$$

$$\leq 2\left[d\log\left(\frac{t}{2d}\right) + \log\frac{1}{\delta}\right]$$

where the second inequality is from $\lambda_{\min}(V_{T_0}) \geq K(1 + P^2)$. Combining these inequalities and setting $\delta = t^{-2}$ gives the result. $\qquad\square$

**Lemma 6.** *Assume that $\|\widehat{\boldsymbol{\theta}}_t - \boldsymbol{\theta}^*\|_{V_t} \leq \alpha_t$ holds for some $\alpha_t$ and let $h_{ti} : \mathbb{R} \to \mathbb{R}$ be the linear function defined as*

$$h_{ti}(p) = \min\{\langle\widehat{\boldsymbol{\psi}}_t, \boldsymbol{x}_{ti}\rangle + g_{ti}, 1\} - \max\{\langle\widehat{\boldsymbol{\phi}}_t, \boldsymbol{x}_{ti}\rangle - g_{ti}, L_0\} \cdot p,$$

*where $\widehat{\boldsymbol{\theta}}_t = (\widehat{\boldsymbol{\psi}}_t, \widehat{\boldsymbol{\phi}}_t)$ and $g_{ti} := \alpha_t\|(\boldsymbol{x}_{ti}, \boldsymbol{x}_{ti})\|_{V_t^{-1}}$. Then, $h_{ti}(p)$ satisfies*

$$h_{ti}(p) \geq u_{ti}(p), \quad \forall p \geq 0 \tag{13}$$

$$h_{ti}(p) - u_{ti}(p) \leq 3Pg_{ti}, \quad \forall p \in [1, P]. \tag{14}$$

*Proof.* Recall the definition of the utility function

$$u_{ti}(p) = \langle\boldsymbol{\psi}^*, \boldsymbol{x}_{ti}\rangle - \langle\boldsymbol{\phi}^*, \boldsymbol{x}_{ti}\rangle \cdot p.$$

Now, to construct the upper bound for $u_{ti}(p)$, we define the function

$$\widetilde{h}_{ti}(p) = \langle\widehat{\boldsymbol{\psi}}_t, \boldsymbol{x}_{ti}\rangle + g_{ti} - (\langle\widehat{\boldsymbol{\phi}}_t, \boldsymbol{x}_{ti}\rangle - g_{ti}) \cdot p$$

and recall the definition $\widetilde{\boldsymbol{x}}_{ti} = (\boldsymbol{x}_{ti}, -p\boldsymbol{x}_{ti})$ to write

$$|\langle\widehat{\boldsymbol{\theta}}_t, \widetilde{\boldsymbol{x}}_{ti}\rangle - \langle\boldsymbol{\theta}^*, \widetilde{\boldsymbol{x}}_{ti}\rangle| = \left|\langle V_t^{1/2}(\widehat{\boldsymbol{\theta}}_t - \boldsymbol{\theta}^*), V_t^{-1/2}\widetilde{\boldsymbol{x}}_{ti}\rangle\right|$$

$$\leq \|V_t^{1/2}(\widehat{\boldsymbol{\theta}}_t - \boldsymbol{\theta}^*)\|\|V_t^{-1/2}\widetilde{\boldsymbol{x}}_{ti}\|$$

$$\leq \|\widehat{\boldsymbol{\theta}}_t - \boldsymbol{\theta}^*\|_{V_t}\|\widetilde{\boldsymbol{x}}_{ti}\|_{V_t^{-1}}$$

$$\leq \|\widehat{\boldsymbol{\theta}}_t - \boldsymbol{\theta}^*\|_{V_t}\|(\boldsymbol{x}_{ti}, -p\boldsymbol{x}_{ti})\|_{V_t^{-1}}$$

Hence,

$$|\langle\widehat{\boldsymbol{\theta}}_t, \widetilde{\boldsymbol{x}}_{ti}\rangle - \langle\boldsymbol{\theta}^*, \widetilde{\boldsymbol{x}}_{ti}\rangle| \leq (1 + p)g_{ti}, \quad \forall p \geq 0$$

$$|\langle\widehat{\boldsymbol{\theta}}_t, \widetilde{\boldsymbol{x}}_{ti}\rangle - \langle\boldsymbol{\theta}^*, \widetilde{\boldsymbol{x}}_{ti}\rangle| \leq Pg_{ti}, \quad \forall p \in [1, P].$$

Then, for all $p \in [1, P]$, we can write

$$h_{ti}(p) - u_{ti}(p) \leq \widetilde{h}_{ti}(p) - u_{ti}(p)$$

$$= \langle\widehat{\boldsymbol{\theta}}_t, \widetilde{\boldsymbol{x}}_{ti}\rangle + (1 + p)g_{ti} - \langle\boldsymbol{\theta}^*, \widetilde{\boldsymbol{x}}_{ti}\rangle$$

$$\leq 3Pg_{ti}$$

where the first inequality follows by noticing $h_{ti}(p) \leq \widetilde{h}_{ti}(p)$ for all $p \geq 0$.

Then, for all $p \geq 0$, we can write

$$u_{ti}(p) - \widetilde{h}_{ti}(p) = \langle\boldsymbol{\theta}^*, \widetilde{\boldsymbol{x}}_{ti}\rangle - \langle\widehat{\boldsymbol{\theta}}_t, \widetilde{\boldsymbol{x}}_{ti}\rangle - (1 + p)g_{ti}$$

$$= \langle\boldsymbol{\psi}^*, \boldsymbol{x}_{ti}\rangle - \langle\widehat{\boldsymbol{\psi}}_t, \boldsymbol{x}_{ti}\rangle - g_{ti} - \left(\langle\boldsymbol{\phi}^*, \boldsymbol{x}_{ti}\rangle - \langle\widehat{\boldsymbol{\phi}}_t, \boldsymbol{x}_{ti}\rangle + g_{ti}\right)p \leq 0$$

which implies that

$$\langle\boldsymbol{\psi}^*, \boldsymbol{x}_{ti}\rangle - \langle\widehat{\boldsymbol{\psi}}_t, \boldsymbol{x}_{ti}\rangle - g_{ti} \leq 0, \text{ and}$$

$$\langle\boldsymbol{\phi}^*, \boldsymbol{x}_{ti}\rangle - \langle\widehat{\boldsymbol{\phi}}_t, \boldsymbol{x}_{ti}\rangle + g_{ti} \geq 0$$

Using the properties of $u_{ti}(p)$ and combining the last two inequalities, we obtain

$$u_{ti}(p) - h_{ti}(p) \leq 0$$

for all $p \geq 0$, proving the result.

$\qquad\square$

**Lemma 7.** *Assume good event $\mathcal{E}$ occurs. Then, for any $t \in [T]$, it holds that*

(a) $R_t(S_t^*, \boldsymbol{p}_t^*) \leq \widetilde{R}_t(S_t, \boldsymbol{p}_t)$, *and*

(b) $\widetilde{R}_t(S_t, \boldsymbol{p}_t) - R_t(S_t, \boldsymbol{p}_t) \leq 3P^2 \max_{i \in S_t} g_{ti}$.

*Proof. Inequality (a):* Fix some $t$ and define revenue functions $R^A : 2^{[n]} \times \mathbb{R}_+^n \to \mathbb{R}$ given by

$$R^A(S, \boldsymbol{p}) = \frac{\sum_{i \in S \setminus A} p_i \exp(u_{ti}(p_i)) + \sum_{i \in S \cap A} p_i \exp(h_{ti}(p_i))}{1 + \sum_{i \in S \setminus A} \exp(u_{ti}(p_i)) + \sum_{i \in S \cap A} \exp(h_{ti}(p_i))}$$

for any $A \subseteq [n]$. Note that this definition leads to $R^\emptyset(S, \boldsymbol{p}) = R_t(S, \boldsymbol{p})$ and $R^S(S, \boldsymbol{p}) = \widetilde{R}_t(S, \boldsymbol{p})$. We also define

$$(S^A, \boldsymbol{p}^A) = \underset{\substack{S \subseteq [n] : |S| \leq K \\ \boldsymbol{p} \in \mathbb{R}_+^n}}{\arg\max} R^A(S, \boldsymbol{p}).$$

which satisfies $(S^\emptyset, \boldsymbol{p}^\emptyset) = (S_t^*, \boldsymbol{p}_t^*)$ and $(S^{[n]}, \boldsymbol{p}^{[n]}) = (S_t, \boldsymbol{p}_t)$.

By the optimality of $(S^A, \boldsymbol{p}^A)$ for any revenue function $R^A$, we have $p_j^A \geq R^A(S^A, \boldsymbol{p}^A)$ for all $j \in S^A$. We can write this inequality as $p_j^A \geq a/b$ where

$$a = \sum_{i \in S^A \setminus A} p_i^A \exp(u_{ti}(p_i^A)) + \sum_{i \in S^A \cap A} p_i^A \exp(h_{ti}(p_i^A)) \quad \text{and}$$
$$b = 1 + \sum_{i \in S^A \setminus A} \exp(u_{ti}(p_i^A)) + \sum_{i \in S^A \cap A} \exp(h_{ti}(p_i^A)).$$

Letting $\delta = \exp(h_{tj}(p_j^A)) - \exp(u_{tj}(p_j^A))$, we have $ab + b\delta p_j^A \geq ab + a\delta$ which implies

$$\frac{a + p_j^A \delta}{b + \delta} \geq \frac{a}{b}.$$

Hence, we have $R^{A \cup \{j\}}(S^A, \boldsymbol{p}^A) \geq R^A(S^A, \boldsymbol{p}^A)$ for all $j \in S^A$.

We also have $R^{A \cup \{j\}}(S^A, \boldsymbol{p}^A) = R^A(S^A, \boldsymbol{p}^A)$ for any $j \notin S^A$. Therefore, $R^{A \cup \{j\}}(S^A, \boldsymbol{p}^A) \geq R^A(S^A, \boldsymbol{p}^A)$ for any $j \in [n]$. Using the optimality of $(S^{A \cup \{j\}}, \boldsymbol{p}^{A \cup \{j\}})$ for function $R^{A \cup \{j\}}$, we can write

$$R^{A \cup \{j\}}(S^{A \cup \{j\}}, \boldsymbol{p}^{A \cup \{j\}}) \geq R^A(S^A, \boldsymbol{p}^A)$$

for any $j \in [n]$. Therefore, by induction, we can show that

$$\widetilde{R}_t(S_t, \boldsymbol{p}_t) = R^{[n]}(S^{[n]}, \boldsymbol{p}^{[n]}) \geq R^\emptyset(S^\emptyset, \boldsymbol{p}^\emptyset) = R_t(S_t^*, \boldsymbol{p}_t^*).$$

*Inequality (b):* Let $u_{ti} := u_{ti}(p_{ti})$ and $h_{ti} := h_{ti}(p_{ti})$ with $3Pg_{ti} \geq h_{ti} - u_{ti} \geq 0$ because $p_{ti} \in [1, P]$. By the mean value theorem, for all $i$, there exists $\bar{u}_{ti} := (1 - c)u_{ti} + ch_{ti}$ for some

$c \in (0, 1)$ with

$$
\begin{aligned}
\widetilde{R}_t(S_t, \boldsymbol{p}_t) - R_t(S_t, \boldsymbol{p}_t) &= \frac{\sum_{i \in S_t} p_{ti} \exp(h_{ti})}{1 + \sum_{j \in S_t} \exp(h_{tj})} - \frac{\sum_{i \in S_t} p_{ti} \exp(u_{ti})}{1 + \sum_{j \in S_t} \exp(u_{tj})} \\
&= \frac{(\sum_{i \in S_t} p_{ti} \exp(\bar{u}_{ti})(h_{ti} - u_{ti}))(1 + \sum_{i \in S_t} \exp(\bar{u}_{ti}))}{(1 + \sum_{i \in S_t} \exp(\bar{u}_{ti}))^2} \\
&\quad - \frac{(\sum_{i \in S_t} p_{ti} \exp(\bar{u}_{ti}))(\sum_{i \in S_t} \exp(\bar{u}_{ti})(h_{ti} - u_{ti}))}{(1 + \sum_{i \in S_t} \exp(\bar{u}_{ti}))^2} \\
&= \sum_{i \in S_t} p_{ti} q_t(i | \bar{\boldsymbol{u}}_t)(h_{ti} - u_{ti}) \\
&\quad - \left( \sum_{i \in S_t} p_{ti} q_t(i | \bar{\boldsymbol{u}}_t) \right) \left( \sum_{i \in S_t} q_t(i | \bar{\boldsymbol{u}}_t)(h_{ti} - u_{ti}) \right) \\
&= \sum_{i \in S_t} \left( p_{ti} - \sum_{i \in S_t} p_{ti} q_t(i | \bar{\boldsymbol{u}}_t) \right) q_t(i | \bar{\boldsymbol{u}}_t)(h_{ti} - u_{ti}) \\
&\leq P \cdot \max_{i \in S_t} (h_{ti} - u_{ti}) \\
&\leq 3 P^2 \max_{i \in S_t} g_{ti}
\end{aligned}
$$

where the first inequality follows from $|p_{ti}| \leq P$ and $q_t(i | \bar{\boldsymbol{u}}_t)$ is a categorical distribution. $\qquad \square$

**Lemma 8.** *Suppose $\|x_{ti}\| \leq 1$ and $1 \leq p_{ti} \leq P$ for all $i$ and $t$. Then $\det(V_t)$ is increasing with respect to $t$ and*

$$
\det(V_t) \leq \left( \frac{tK(1 + P^2)}{2d} \right)^{2d}
$$

*Proof.* Let $\lambda_1, \ldots, \lambda_{2d}$ be the eigenvalues of $V_t$. Then, using the AM-GM inequality we can write

$$
\begin{aligned}
\det(V_t) &= \prod_{i=1}^{2d} \lambda_i \\
&\leq \left( \frac{\sum_{i=1}^{2d} \lambda_i}{2d} \right)^{2d} \\
&= \left( \frac{\operatorname{trace}(V_t)}{2d} \right)^{2d} \\
&= \left( \frac{\sum_{\tau=1}^{t} \sum_{i \in S_\tau} \|\widetilde{\boldsymbol{x}}_{\tau i}\|_2^2}{2d} \right)^{2d} \\
&\leq \left( \frac{tK(1 + P^2)}{2d} \right)^{2d}.
\end{aligned}
$$

$\qquad \square$

**Lemma 9.** *Suppose $\|x_{ti}\| \leq 1$ and $p_{ti} \leq P$ for all $i$ and $t$. If $\lambda_{\min}(V_{T_0}) \geq 2PK$, then*

$$
\sum_{t=T_0+1}^{n} \sum_{i \in S_t} \|\widetilde{\boldsymbol{x}}_{ti}\|_{V_t^{-1}}^2 \leq 2 \log \left( \frac{\det(V_n)}{\det(V_{T_0})} \right)
$$

*Proof.* Let $\lambda_1, \ldots, \lambda_{2d}$ be the eigenvalues of $\sum_{i \in S_t} \widetilde{\boldsymbol{x}}_{ti} \widetilde{\boldsymbol{x}}_{ti}^\top$. Since $\sum_{i \in S_t} \widetilde{\boldsymbol{x}}_{ti} \widetilde{\boldsymbol{x}}_{ti}^\top$ is positive semi-definite, $\lambda_j \geq 0$ for all $j$. Then, we have

$$
\begin{aligned}
\det\left( \boldsymbol{I} + \sum_{i \in S_t} \widetilde{\boldsymbol{x}}_{ti} \widetilde{\boldsymbol{x}}_{ti}^\top \right) &= \prod_{i=1}^{2d} (1 + \lambda_j) \\
&\geq 1 + \sum_{i=1}^{2d} \lambda_j \\
&= 1 - 2d + \sum_{i=1}^{2d} (1 + \lambda_j) \\
&= 1 - 2d + \operatorname{trace}\left( \boldsymbol{I} + \sum_{i \in S_t} \widetilde{\boldsymbol{x}}_{ti} \widetilde{\boldsymbol{x}}_{ti}^\top \right) \\
&= 1 + \sum_{i \in S_t} \|\widetilde{\boldsymbol{x}}_{ti}\|_2^2
\end{aligned}
$$

Now, to lower bound $\det(V_{n+1})$, we write

$$
\begin{aligned}
\det(V_{n+1}) &= \det\left( V_n + \sum_{i \in S_n} \widetilde{\boldsymbol{x}}_{ti} \widetilde{\boldsymbol{x}}_{ti}^\top \right) \\
&= \det(V_n) \left( \boldsymbol{I} + \sum_{i \in S_n} V_n^{-1/2} \widetilde{\boldsymbol{x}}_{ti} (V_n^{-1/2} \widetilde{\boldsymbol{x}}_{ti})^\top \right) \\
&\geq \det(V_n) \left( 1 + \sum_{i \in S_n} \|\widetilde{\boldsymbol{x}}_{ti}\|_{V_n^{-1}}^2 \right) \\
&\geq \det(V_{T_0}) \prod_{t=T_0}^{n} \left( 1 + \sum_{i \in S_t} \|\widetilde{\boldsymbol{x}}_{ti}\|_{V_t^{-1}}^2 \right)
\end{aligned}
$$

Notice that

$$
\sum_{i \in S_t} \|\widetilde{\boldsymbol{x}}_{ti}\|_{V_t^{-1}}^2 \leq \frac{\|\widetilde{\boldsymbol{x}}_{ti}\|^2}{\lambda_{\min}(V_t)} \leq \frac{(1+P)}{\lambda_{\min}(V_t)} \leq \frac{(1+P)}{2PK} \leq \frac{1}{K}.
$$

Hence, $\sum_{i \in S_t} \|\widetilde{\boldsymbol{x}}_{ti}\|_{V_t^{-1}}^2 \leq 1$ for all $t \geq T_0$. Then, using the fact that $z \leq 2\log(1+z)$ for any $z \in [0, 1]$, we have

$$
\begin{aligned}
\sum_{t=T_0+1}^{n} \sum_{i \in S_t} \|\widetilde{\boldsymbol{x}}_{ti}\|_{V_t^{-1}}^2 &\leq 2 \sum_{t=T_0+1}^{n} \log\left( 1 + \sum_{i \in S_t} \|\widetilde{\boldsymbol{x}}_{ti}\|_{V_t^{-1}}^2 \right) \\
&= 2\log \prod_{t=T_0+1}^{n} \left( 1 + \sum_{i \in S_t} \|\widetilde{\boldsymbol{x}}_{ti}\|_{V_t^{-1}}^2 \right) \\
&\leq 2\log\left( \frac{\det(V_n)}{\det(V_{T_0})} \right)
\end{aligned}
$$

$\square$

**Lemma 10.** *If $\lambda_{\min}(V_{T_0}) \geq K(1 + P^2)$, then we have*

$$
\sum_{t=1}^{T} \max_{i \in S_t} \|\widetilde{\boldsymbol{x}}_{ti}\|_{V_t^{-1}}^2 \leq 4d\log(T/d).
$$

*Noting that $p_{ti} \geq 1$ for all $i$ and $t$, we also have*

$$\sum_{t=1}^{T} \max_{i \in S_t} \|(\boldsymbol{x}_{ti}, \boldsymbol{x}_{ti})\|_{V_t^{-1}}^2 \leq 4d \log(T/d).$$

*Proof.* Combining Lemma 8 and Lemma 9, we obtain

$$\sum_{t=1}^{T} \max_{i \in S_t} \|\widetilde{\boldsymbol{x}}_{ti}\|_{V_t^{-1}}^2 \leq 2 \log\left(\frac{\det(V_T)}{\det(V_{T_0})}\right) \leq 2 \log\left(\frac{TK(1+P^2)}{2d\lambda_{\min}(V_{T_0})}\right)^{2d} \leq 4d \log(T/d)$$

where the last inequality is by $\lambda_{\min}(V_{T_0}) \geq K(1+P^2)$. $\qquad\square$

## C   PROOF OF THEOREM 4

Similar to Algorithm 2, we run $T_0$ initialization round with random assortment and price selections to obtain an initial MLE $\boldsymbol{\theta}_0 := \widehat{\boldsymbol{\theta}}_{T_0}$. Using the results of Lemma 2 and Lemma 4, we can show that the condition $\|\boldsymbol{\theta}_0 - \boldsymbol{\theta}^*\| \leq 1/2$ is satisfied with probability $1 - \mathcal{O}(T^{-2})$ if we select

$$T_0 = \Theta\left(\frac{d + \log T + P^2}{\sigma_0^2 \kappa^2}\right). \tag{16}$$

Then, we apply the following parameter update at each time step $t$.

$$\widehat{\boldsymbol{\theta}}_t = \underset{\boldsymbol{\theta}:\|\boldsymbol{\theta}-\boldsymbol{\theta}_0\|\leq 1/2}{\arg\min} \left\{\|\boldsymbol{\theta} - \widehat{\boldsymbol{\theta}}_{t-1}\|_{V_t}^2 + \frac{4}{\kappa}(\boldsymbol{\theta} - \widehat{\boldsymbol{\theta}}_{t-1})^\top g_t(\widehat{\boldsymbol{\theta}}_{t-1})\right\} \tag{17}$$

which ensures that $\|\widehat{\boldsymbol{\theta}}_t - \boldsymbol{\theta}^*\| \leq 1$ for all $t \geq T_0$ with probability $1 - \mathcal{O}(T^{-2})$. For this parameter update, only the $\Theta(K)$ context vector in the last offered assortment is needed per each round, compared to $\Theta(tK)$ in Algorithm 2 which grows linearly with each round $t$. Then, using this update rule, we modify our algorithm and present it in the next algorithm block.

---

**Algorithm 3** DASP-MNL with online parameter updates

---

1: **Input:** initialization rounds $T_0$, confidence parameters $\{\alpha_t\}_{t\in[T]}$, minimum price sensitivity $L_0$
2: $V_0 \leftarrow \mathbf{0} \in \mathbb{R}^{2d \times 2d}$
3: **for** $t = 1, 2, \ldots, T_0$ **do**                               ▷ initialization rounds
4:     Choose $S_t$ uniformly at random from $\{S \subseteq [n] : |S| \leq K\}$
5:     Choose $p_{ti}$ independently and uniformly at random from $[1, 2]$ for all $i \in S_t$
6:     Offer assortment $S_t$ at price $\boldsymbol{p}_t$ and observe $i_t$
7:     $V_t \leftarrow V_{t-1} + \sum_{i \in S_t} \widetilde{\boldsymbol{x}}_{ti}\widetilde{\boldsymbol{x}}_{ti}^\top$
8: Compute MLE $\widehat{\boldsymbol{\theta}}_{T_0}$ by solving equation 6 and set $\boldsymbol{\theta}_0 = \widehat{\boldsymbol{\theta}}_{T_0}$.
9: **for** $t = T_0 + 1, T_0 + 2, \ldots, T$ **do**
10:     Compute $g_{ti} := \alpha_t \|(\boldsymbol{x}_{ti}, \boldsymbol{x}_{ti})\|_{V_t^{-1}}$ for all $i \in [n]$          ▷ Confidence bonus
11:     Compute $\widehat{\boldsymbol{\theta}}_t$ by solving equation 17                        ▷ Online parameter update
12:     Let $h_{ti}(p) = \min\{\langle\widehat{\boldsymbol{\psi}}_t, \boldsymbol{x}_{ti}\rangle + g_{ti}, 1\} - \max\{\langle\widehat{\boldsymbol{\phi}}_t, \boldsymbol{x}_{ti}\rangle - g_{ti}, L_0\} \cdot p$ for all $i \in [n]$
13:     Choose $(S_t, \boldsymbol{p}_t)$ using Algorithm 1 with linear functions $h_{ti}(p)$
14:     Offer assortment $S_t$ at price $\boldsymbol{p}_t$ and observe $i_t$
15:     $V_t \leftarrow V_{t-1} + \sum_{i \in S_t} \widetilde{\boldsymbol{x}}_{ti}\widetilde{\boldsymbol{x}}_{ti}^\top$

---

To analyze the regret of Algorithm 3, we first define a per-round negative log-likelihood function $f_t(\boldsymbol{\theta})$ and its gradient $\nabla_{\boldsymbol{\theta}} f_t(\boldsymbol{\theta})$ as

$$f_t(\boldsymbol{\theta}) = -q_t(i_t|S_t, \boldsymbol{p}_t; \boldsymbol{\theta})$$

$$g_t(\boldsymbol{\theta}) = \nabla_{\boldsymbol{\theta}} f_t(\boldsymbol{\theta}) = \sum_{i \in S_t} q_t(i|S_t, \boldsymbol{p}_t; \boldsymbol{\theta})\widetilde{\boldsymbol{x}}_{ti} - \widetilde{\boldsymbol{x}}_{ti_t}.$$

We note that negative log-likelihood $f_t(\boldsymbol{\theta})$ for MNL model at each round $t$ is a strongly convex function over a bounded domain, which enables us to apply a variant of online Newton updates (Hazan et al., 2014) that was also used in Hazan et al. (2014); Zhang et al. (2016); Oh & Iyengar (2021) which proposed online algorithms for the logistic models.

To prove the regret rate for our algorithm with online parameter updates, we construct a new utility function upper-bound estimate $\widetilde{h}_{ti}(p)$ using a new confidence radius $\widetilde{\alpha}_t$ specified in the following lemma.

**Lemma 11.** *Let $T_0$ be any round such that $\lambda_{\min}(V_{T_0}) \geq KP^2$. Then, for any $t > T_0$, we have $\|\widehat{\boldsymbol{\theta}}_t - \boldsymbol{\theta}^*\|_{V_t} \leq \widetilde{\alpha}_t$ with probability at least $1 - t^{-2}$ for confidence radius*

$$\widetilde{\alpha}_t = \sqrt{T_0 + \frac{64}{\kappa^2} d \log\left(\frac{t}{d}\right) + \frac{4}{\kappa}\left(\frac{4}{\kappa} + \frac{8}{3}\right) \log\left(\lceil 2\log_2(tK/2)\rceil t^4\right) + \frac{8}{\kappa} + 1}. \qquad (18)$$

Then, similar to the proof of Theorem 3, we define a *good* event $\widetilde{\mathcal{E}}_t = \{\|\widehat{\boldsymbol{\theta}}_t - \boldsymbol{\theta}^*\|_{V_t} \leq \widetilde{\alpha}_t\}$ for $t \geq T_0$ that holds with probability at least $1 - t^{-2}$. Consequently, following steps similar to the proof of Theorem 3, we can write the regret as

$$\mathcal{R}_T \leq PT_0 + \sum_{t=T_0+1}^{T} \mathbb{E}\left[\left(\widetilde{R}_t(S_t, \boldsymbol{p}_t) - R_t(S_t, \boldsymbol{p}_t)\right)\mathbf{1}(\mathcal{E}_t)\right] + \mathbb{E}\left[\left(\widetilde{R}_t(S_t, \boldsymbol{p}_t) - R_t(S_t, \boldsymbol{p}_t)\right)\mathbf{1}(\neg\mathcal{E}_t)\right]$$

$$\leq PT_0 + \sum_{t=T_0+1}^{T} 3P^2\alpha_t \max_{i \in S_t}\|(\boldsymbol{x}_{ti}, \boldsymbol{x}_{ti})\|_{V_t^{-1}} + \mathcal{O}(P)$$

Finally, using Lemma 10, we show that

$$\mathcal{R}_T \leq PT_0 + 6P^2\widetilde{\alpha}_T\sqrt{Td\log(T/d)} + \mathcal{O}(P).$$

for $\widetilde{\alpha}_T$ given in Lemma 11. Note that $\widetilde{\alpha}_T = \mathcal{O}(\sqrt{d\log(T)})$ for the selection of $T_0$ given in equation 10.

## C.1 PROOF OF LEMMA 11

The proof of Lemma 11 depends on a few technical results we present next. First, we define the matrix $W_t = \sum_{i \in S_t} \widetilde{\boldsymbol{x}}_{\tau i}\widetilde{\boldsymbol{x}}_{\tau i}^\top$.

We start by showing that following bound holds true over $\mathcal{B} := \{\boldsymbol{\theta} : \|\boldsymbol{\theta} - \boldsymbol{\theta}^*\| \leq 1\}$.

**Lemma 12.** *For any $\boldsymbol{\theta}_1, \boldsymbol{\theta}_2 \in \mathcal{B}$, we have*

$$f_t(\boldsymbol{\theta}_2) \geq f_t(\boldsymbol{\theta}_1) + g_t(\boldsymbol{\theta}_1)^\top(\boldsymbol{\theta}_2 - \boldsymbol{\theta}_1) + \frac{\kappa}{2}(\boldsymbol{\theta}_2 - \boldsymbol{\theta}_1)^\top W_t(\boldsymbol{\theta}_2 - \boldsymbol{\theta}_1).$$

*Proof.* Using the Taylor's expansion, there exists some $c \in [0, 1]$ such that

$$f_t(\boldsymbol{\theta}_2) = f_t(\boldsymbol{\theta}_1) + g_t(\boldsymbol{\theta}_1)^\top(\boldsymbol{\theta}_2 - \boldsymbol{\theta}_1) + (\boldsymbol{\theta}_2 - \boldsymbol{\theta}_1)^\top H(\overline{\boldsymbol{\theta}})(\boldsymbol{\theta}_2 - \boldsymbol{\theta}_1)$$

where $\overline{\boldsymbol{\theta}} = c\boldsymbol{\theta}_2 + (1-c)\boldsymbol{\theta}_1$ and $H(\overline{\boldsymbol{\theta}})$ is the Hessian of $f_t$ at $\overline{\boldsymbol{\theta}}$.

Following the proof of Proposition 2, the Hessian matrix can be lower bounded as

$$H(\overline{\boldsymbol{\theta}}) = \sum_{i \in S_t} q_{ti}(\boldsymbol{\theta})\widetilde{\boldsymbol{x}}_{ti}\widetilde{\boldsymbol{x}}_{ti}^\top - \sum_{i \in S_t}\sum_{j \in S_t} q_{ti}(\boldsymbol{\theta})q_{tj}(\boldsymbol{\theta})\widetilde{\boldsymbol{x}}_{ti}\widetilde{\boldsymbol{x}}_{tj}^\top$$

$$\succcurlyeq \sum_{i \in S_t} q_{ti}(\boldsymbol{\theta})q_{t0}(\boldsymbol{\theta})\widetilde{\boldsymbol{x}}_{ti}\widetilde{\boldsymbol{x}}_{ti}^\top$$

$$\succcurlyeq \kappa W_t$$

where the last step follows from Lemma 3. Consequently, the result follows. $\qquad \square$

Next, we prove the following lemma that shows the dependency between the error $(\widehat{\boldsymbol{\theta}}_t - \boldsymbol{\theta}^*)$ at time $t$ and the error $(\widehat{\boldsymbol{\theta}}_{t+1} - \boldsymbol{\theta}^*)$ at time $t + 1$.

**Lemma 13.** *For any $t$,*

$$g_t(\widehat{\boldsymbol{\theta}}_t)^\top (\widehat{\boldsymbol{\theta}}_t - \boldsymbol{\theta}^*) \leq \frac{1}{\kappa} \|g_t(\widehat{\boldsymbol{\theta}}_t)\|_{V_{t+1}^{-1}}^2 + \frac{\kappa}{4} \|\widehat{\boldsymbol{\theta}}_t - \boldsymbol{\theta}^*\|_{V_{t+1}}^2 - \frac{\kappa}{4} \|\widehat{\boldsymbol{\theta}}_{t+1} - \boldsymbol{\theta}^*\|_{V_{t+1}}^2.$$

*Proof.* Note that

$$\widehat{\boldsymbol{\theta}}_{t+1} = \underset{\boldsymbol{\theta}:\|\boldsymbol{\theta}-\boldsymbol{\theta}_0\|\leq 1/2}{\arg\min} \left\{ \frac{1}{2} \|\boldsymbol{\theta} - \widehat{\boldsymbol{\theta}}_t\|_{V_{t+1}}^2 + \frac{2}{\kappa} (\boldsymbol{\theta} - \widehat{\boldsymbol{\theta}}_t)^\top g_t(\widehat{\boldsymbol{\theta}}_t) \right\}.$$

Hence, from the first-order optimality condition, we have

$$\left( \frac{2}{\kappa} g_t(\widehat{\boldsymbol{\theta}}_t) + V_{t+1}(\widehat{\boldsymbol{\theta}}_{t+1} - \widehat{\boldsymbol{\theta}}_t) \right)^\top (\boldsymbol{\theta} - \widehat{\boldsymbol{\theta}}_{t+1}) \geq 0$$

for any $\boldsymbol{\theta}$ such that $\|\boldsymbol{\theta} - \boldsymbol{\theta}_0\| \leq 1/2$. We can rewrite this inequality as

$$\boldsymbol{\theta}^\top V_{t+1}(\widehat{\boldsymbol{\theta}}_{t+1} - \widehat{\boldsymbol{\theta}}_t) \geq \widehat{\boldsymbol{\theta}}_{t+1}^\top V_{t+1}(\widehat{\boldsymbol{\theta}}_{t+1} - \widehat{\boldsymbol{\theta}}_t) - \frac{2}{\kappa} g_t(\widehat{\boldsymbol{\theta}}_t)^\top (\boldsymbol{\theta} - \widehat{\boldsymbol{\theta}}_{t+1}).$$

Then, we can write

$$\begin{aligned}
&\|\widehat{\boldsymbol{\theta}}_t - \boldsymbol{\theta}^*\|_{V_{t+1}}^2 - \|\widehat{\boldsymbol{\theta}}_{t+1} - \boldsymbol{\theta}^*\|_{V_{t+1}}^2 \\
&= \widehat{\boldsymbol{\theta}}_t^\top V_{t+1} \widehat{\boldsymbol{\theta}}_t - \widehat{\boldsymbol{\theta}}_{t+1}^\top V_{t+1} \widehat{\boldsymbol{\theta}}_{t+1} + 2\boldsymbol{\theta}^{*\top} V_{t+1}(\widehat{\boldsymbol{\theta}}_{t+1} - \widehat{\boldsymbol{\theta}}_t) \\
&\geq \widehat{\boldsymbol{\theta}}_t^\top V_{t+1} \widehat{\boldsymbol{\theta}}_t - \widehat{\boldsymbol{\theta}}_{t+1}^\top V_{t+1} \widehat{\boldsymbol{\theta}}_{t+1} + 2\widehat{\boldsymbol{\theta}}_{t+1}^\top V_{t+1}(\widehat{\boldsymbol{\theta}}_{t+1} - \widehat{\boldsymbol{\theta}}_t) - \frac{4}{\kappa} g_t(\widehat{\boldsymbol{\theta}}_t)^\top (\boldsymbol{\theta}^* - \widehat{\boldsymbol{\theta}}_{t+1}) \\
&= \widehat{\boldsymbol{\theta}}_t^\top V_{t+1} \widehat{\boldsymbol{\theta}}_t + \widehat{\boldsymbol{\theta}}_{t+1}^\top V_{t+1} \widehat{\boldsymbol{\theta}}_{t+1} - 2\widehat{\boldsymbol{\theta}}_{t+1}^\top V_{t+1} \widehat{\boldsymbol{\theta}}_t - \frac{4}{\kappa} g_t(\widehat{\boldsymbol{\theta}}_t)^\top (\boldsymbol{\theta}^* - \widehat{\boldsymbol{\theta}}_{t+1}) \\
&= \|\widehat{\boldsymbol{\theta}}_{t+1} - \widehat{\boldsymbol{\theta}}_t\|_{V_{t+1}}^2 + \frac{4}{\kappa} g_t(\widehat{\boldsymbol{\theta}}_t)^\top (\widehat{\boldsymbol{\theta}}_{t+1} - \widehat{\boldsymbol{\theta}}_t) + \frac{4}{\kappa} g_t(\widehat{\boldsymbol{\theta}}_t)^\top (\widehat{\boldsymbol{\theta}}_t - \boldsymbol{\theta}^*) \\
&\geq -\frac{4}{\kappa^2} \|g_t(\widehat{\boldsymbol{\theta}}_t)\|_{V_{t+1}^{-1}}^2 + \frac{4}{\kappa} g_t(\widehat{\boldsymbol{\theta}}_t)^\top (\widehat{\boldsymbol{\theta}}_t - \boldsymbol{\theta}^*)
\end{aligned}$$

where the last inequality follows from

$$\begin{aligned}
\|\widehat{\boldsymbol{\theta}}_{t+1} - \widehat{\boldsymbol{\theta}}_t\|_{V_{t+1}}^2 + \frac{4}{\kappa} g_t(\widehat{\boldsymbol{\theta}}_t)^\top (\widehat{\boldsymbol{\theta}}_{t+1} - \widehat{\boldsymbol{\theta}}_t) &\geq \underset{\boldsymbol{\theta}:\|\boldsymbol{\theta}-\boldsymbol{\theta}_0\|\leq 1/2}{\min} \left\{ \|\boldsymbol{\theta} - \widehat{\boldsymbol{\theta}}_t\|_{V_{t+1}}^2 + \frac{4}{\kappa} (\boldsymbol{\theta} - \widehat{\boldsymbol{\theta}}_t)^\top g_t(\widehat{\boldsymbol{\theta}}_t) \right\} \\
&\geq \underset{\boldsymbol{\theta}}{\min} \left\{ \|\boldsymbol{\theta} - \widehat{\boldsymbol{\theta}}_t\|_{V_{t+1}}^2 + \frac{4}{\kappa} (\boldsymbol{\theta} - \widehat{\boldsymbol{\theta}}_t)^\top g_t(\widehat{\boldsymbol{\theta}}_t) \right\} \\
&= -\frac{4}{\kappa^2} \|g_t(\widehat{\boldsymbol{\theta}}_t)\|_{V_{t+1}^{-1}}^2
\end{aligned}$$

$\square$

**Lemma 14.** *For any positive definite matrix $V$,*

$$\|g_t(\boldsymbol{\theta})\|_V^2 \leq 4 \max_{i \in S_t} \|\widetilde{\boldsymbol{x}}_{ti}\|_V^2.$$

*Proof.* Let $y_{ti}$ be a binary variable such that $y_{ti} = 1$ if $i_t = i$ and $y_{ti} = 0$ otherwise. For convenience also denote $q_{ti} = q_t(i|S_t, \boldsymbol{p}_t; \boldsymbol{\theta})$. Then, we note that $\sum_{i \in S_t} q_{ti} \leq 1$ and $\sum_{i \in S_t} y_{ti} \leq 1$.

Consequently, we can write

$$\|g_t(\boldsymbol{\theta})\|_V^2 = \sum_{i \in S_t} \sum_{j \in S_t} (q_{ti} - y_{ti})(q_{tj} - y_{tj}) \, \widetilde{\boldsymbol{x}}_{ti}^\top V \widetilde{\boldsymbol{x}}_{tj}$$

$$\leq \sum_{i \in S_t} \sum_{j \in S_t} (q_{ti}q_{tj} + y_{ti}y_{tj} + q_{ti}y_{tj} + q_{tj}y_{ti}) \, |\widetilde{\boldsymbol{x}}_{ti}^\top V \widetilde{\boldsymbol{x}}_{tj}|$$

$$\leq 4 \max_{i,j \in S_t} |\widetilde{\boldsymbol{x}}_{ti}^\top V \widetilde{\boldsymbol{x}}_{tj}|$$

$$\leq 4 \max_{i \in S_t} |\widetilde{\boldsymbol{x}}_{ti}^\top V \widetilde{\boldsymbol{x}}_{ti}|$$

$$= 4 \max_{i \in S_t} \|\widetilde{\boldsymbol{x}}_{ti}\|_V^2.$$

$\square$

Next, we let $\mathcal{F}_t$ denote the filtration up to time $t$ and define the conditional expected values for the per-round negative log-likelihood $f_t(\boldsymbol{\theta})$ and its gradient $g_t(\boldsymbol{\theta})$ as follows.

$$\bar{f}_t(\boldsymbol{\theta}) = \mathbb{E}_{i_t}[f_t(\boldsymbol{\theta})|\mathcal{F}_t]$$
$$\bar{g}_t(\boldsymbol{\theta}) = \mathbb{E}_{i_t}[g_t(\boldsymbol{\theta})|\mathcal{F}_t].$$

Then, we show that $\bar{f}_t(\boldsymbol{\theta})$ is minimized at $\boldsymbol{\theta}^*$. Formally, we prove the following lemma.

**Lemma 15.** *For any $\boldsymbol{\theta} \in \mathbb{R}^{2d}$, we have $\bar{f}_t(\boldsymbol{\theta}) \geq \bar{f}_t(\boldsymbol{\theta}^*)$.*

*Proof.* For any $\boldsymbol{\theta} \in \mathbb{R}^{2d}$,

$$\bar{f}_t(\boldsymbol{\theta}) - \bar{f}_t(\boldsymbol{\theta}^*) = \sum_{i \in S_t} q_t(i|S_t, \boldsymbol{p}_t; \boldsymbol{\theta}^*)[\log q_t(i|S_t, \boldsymbol{p}_t; \boldsymbol{\theta}^*) - \log q_t(i|S_t, \boldsymbol{p}_t; \boldsymbol{\theta})]$$

$$\geq 0$$

since it is equal to the Kullback-Leibler (KL) divergence between distributions $q_t(i|S_t, \boldsymbol{p}_t; \boldsymbol{\theta}^*)$ and $q_t(i|S_t, \boldsymbol{p}_t; \boldsymbol{\theta})$. $\square$

**Lemma 16** (Lemma 14 of Oh & Iyengar (2021)). *With probability at least $1 - \delta$,*

$$\sum_{\tau=T_0+1}^{t} \left( \bar{g}_\tau(\widehat{\boldsymbol{\theta}}_\tau) - g_\tau(\widehat{\boldsymbol{\theta}}_\tau) \right)^\top (\widehat{\boldsymbol{\theta}}_\tau - \boldsymbol{\theta}^*)$$

$$\leq \frac{\kappa}{4} \sum_{\tau=T_0+1}^{t} \|\widehat{\boldsymbol{\theta}}_\tau - \boldsymbol{\theta}^*\|_{W_\tau}^2 + \left( \frac{4}{\kappa} + \frac{8}{3} \right) \log \left( \frac{\lceil 2 \log_2 \frac{tK}{2} \rceil t^2}{\delta} \right) + 2.$$

Now, we prove Lemma 11 by using the previous results. First, we note that $\widehat{\boldsymbol{\theta}}_t, \boldsymbol{\theta}^* \in \mathcal{B}$ for $t \geq T_0$ and use Lemma 12 to write

$$f_t(\widehat{\boldsymbol{\theta}}_t) \leq f_t(\boldsymbol{\theta}^*) + g_t(\widehat{\boldsymbol{\theta}}_t)^\top (\widehat{\boldsymbol{\theta}}_t - \boldsymbol{\theta}^*) - \frac{\kappa}{2} (\widehat{\boldsymbol{\theta}}_t - \boldsymbol{\theta}^*)^\top W_t (\widehat{\boldsymbol{\theta}}_t - \boldsymbol{\theta}^*).$$

Then, by taking the expectation over $i_t$ on both sides, we obtain

$$\bar{f}_t(\widehat{\boldsymbol{\theta}}_t) \leq \bar{f}_t(\boldsymbol{\theta}^*) + \bar{g}_t(\widehat{\boldsymbol{\theta}}_t)^\top (\widehat{\boldsymbol{\theta}}_t - \boldsymbol{\theta}^*) - \frac{\kappa}{2} (\widehat{\boldsymbol{\theta}}_t - \boldsymbol{\theta}^*)^\top W_t (\widehat{\boldsymbol{\theta}}_t - \boldsymbol{\theta}^*).$$

Since $\bar{f}_t(\boldsymbol{\theta}) \geq \bar{f}_t(\boldsymbol{\theta}^*)$ by Lemma 15, we have

$$0 \leq \bar{f}_t(\widehat{\boldsymbol{\theta}}_t) - \bar{f}_t(\boldsymbol{\theta}^*)$$

$$\leq \bar{g}_t(\widehat{\boldsymbol{\theta}}_t)^\top (\widehat{\boldsymbol{\theta}}_t - \boldsymbol{\theta}^*) - \frac{\kappa}{2} \|\widehat{\boldsymbol{\theta}}_t - \boldsymbol{\theta}^*\|_{W_t}^2$$

$$\leq g_t(\widehat{\boldsymbol{\theta}}_t)^\top (\widehat{\boldsymbol{\theta}}_t - \boldsymbol{\theta}^*) - \frac{\kappa}{2} \|\widehat{\boldsymbol{\theta}}_t - \boldsymbol{\theta}^*\|_{W_t}^2 + \left( \bar{g}_t(\widehat{\boldsymbol{\theta}}_t) - g_t(\widehat{\boldsymbol{\theta}}_t) \right)^\top (\widehat{\boldsymbol{\theta}}_t - \boldsymbol{\theta}^*).$$

Using Lemma 13 and Lemma 14, we have

$$
\begin{aligned}
0 \le\ & \frac{1}{\kappa}\|g_t(\widehat{\boldsymbol{\theta}}_t)\|_{V_{t+1}^{-1}}^2 + \frac{\kappa}{4}\|\widehat{\boldsymbol{\theta}}_t - \boldsymbol{\theta}^*\|_{V_{t+1}}^2 - \frac{\kappa}{4}\|\widehat{\boldsymbol{\theta}}_{t+1} - \boldsymbol{\theta}^*\|_{V_{t+1}}^2 \\
& - \frac{\kappa}{2}\|\widehat{\boldsymbol{\theta}}_t - \boldsymbol{\theta}^*\|_{W_t}^2 + \left(\bar{g}_t(\widehat{\boldsymbol{\theta}}_t) - g_t(\widehat{\boldsymbol{\theta}}_t)\right)^\top (\widehat{\boldsymbol{\theta}}_t - \boldsymbol{\theta}^*) \\
\le\ & \frac{4}{\kappa} \max_{i \in S_t} \|\widetilde{\boldsymbol{x}}_{ti}\|_{V_{t+1}^{-1}}^2 + \frac{\kappa}{4}\|\widehat{\boldsymbol{\theta}}_t - \boldsymbol{\theta}^*\|_{V_{t+1}}^2 - \frac{\kappa}{4}\|\widehat{\boldsymbol{\theta}}_{t+1} - \boldsymbol{\theta}^*\|_{V_{t+1}}^2 \\
& - \frac{\kappa}{2}\|\widehat{\boldsymbol{\theta}}_t - \boldsymbol{\theta}^*\|_{W_t}^2 + \left(\bar{g}_t(\widehat{\boldsymbol{\theta}}_t) - g_t(\widehat{\boldsymbol{\theta}}_t)\right)^\top (\widehat{\boldsymbol{\theta}}_t - \boldsymbol{\theta}^*) \\
\le\ & \frac{4}{\kappa} \max_{i \in S_t} \|\widetilde{\boldsymbol{x}}_{ti}\|_{V_{t+1}^{-1}}^2 + \frac{\kappa}{4}\|\widehat{\boldsymbol{\theta}}_t - \boldsymbol{\theta}^*\|_{V_t}^2 - \frac{\kappa}{4}\|\widehat{\boldsymbol{\theta}}_{t+1} - \boldsymbol{\theta}^*\|_{V_{t+1}}^2 \\
& - \frac{\kappa}{4}\|\widehat{\boldsymbol{\theta}}_t - \boldsymbol{\theta}^*\|_{W_t}^2 + \left(\bar{g}_t(\widehat{\boldsymbol{\theta}}_t) - g_t(\widehat{\boldsymbol{\theta}}_t)\right)^\top (\widehat{\boldsymbol{\theta}}_t - \boldsymbol{\theta}^*).
\end{aligned}
$$

where the last inequality follows by noting that we have

$$
\|\widehat{\boldsymbol{\theta}}_t - \boldsymbol{\theta}^*\|_{V_{t+1}}^2 = \|\widehat{\boldsymbol{\theta}}_t - \boldsymbol{\theta}^*\|_{V_t}^2 + \|\widehat{\boldsymbol{\theta}}_t - \boldsymbol{\theta}^*\|_{W_t}^2.
$$

for $V_{t+1} = V_t + W_t$.

Hence, we have

$$
\begin{aligned}
\|\widehat{\boldsymbol{\theta}}_{t+1} - \boldsymbol{\theta}^*\|_{V_{t+1}}^2 \le\ & \|\widehat{\boldsymbol{\theta}}_t - \boldsymbol{\theta}^*\|_{V_t}^2 + \frac{16}{\kappa^2} \max_{i \in S_t} \|\widetilde{\boldsymbol{x}}_{ti}\|_{V_{t+1}^{-1}}^2 - \|\widehat{\boldsymbol{\theta}}_t - \boldsymbol{\theta}^*\|_{W_t}^2 \\
& + \frac{4}{\kappa} \left(\bar{g}_t(\widehat{\boldsymbol{\theta}}_t) - g_t(\widehat{\boldsymbol{\theta}}_t)\right)^\top (\widehat{\boldsymbol{\theta}}_t - \boldsymbol{\theta}^*).
\end{aligned}
$$

Summing over $\tau = T_0 + 1, \ldots, t$, we obtain

$$
\begin{aligned}
\|\widehat{\boldsymbol{\theta}}_{t+1} - \boldsymbol{\theta}^*\|_{V_{t+1}}^2 \le\ & \|\widehat{\boldsymbol{\theta}}_t - \boldsymbol{\theta}^*\|_{V_{T_0+1}}^2 + \frac{16}{\kappa^2} \sum_{\tau=T_0+1}^t \max_{i \in S_\tau} \|\widetilde{\boldsymbol{x}}_{\tau i}\|_{V_{\tau+1}^{-1}}^2 - \sum_{\tau=T_0+1}^t \|\widehat{\boldsymbol{\theta}}_\tau - \boldsymbol{\theta}^*\|_{W_\tau}^2 \\
& + \frac{4}{\kappa} \sum_{\tau=T_0+1}^t \left(\bar{g}_\tau(\widehat{\boldsymbol{\theta}}_\tau) - g_\tau(\widehat{\boldsymbol{\theta}}_\tau)\right)^\top (\widehat{\boldsymbol{\theta}}_\tau - \boldsymbol{\theta}^*).
\end{aligned}
$$

Then, Lemma 16 shows with a probability at least $1 - \delta$,

$$
\begin{aligned}
& \|\widehat{\boldsymbol{\theta}}_{t+1} - \boldsymbol{\theta}^*\|_{V_{t+1}}^2 \\
& \le \|\widehat{\boldsymbol{\theta}}_t - \boldsymbol{\theta}^*\|_{V_{T_0+1}}^2 + \frac{16}{\kappa^2} \sum_{\tau=T_0+1}^t \max_{i \in S_\tau} \|\widetilde{\boldsymbol{x}}_{\tau i}\|_{V_{\tau+1}^{-1}}^2 + \frac{4}{\kappa}\left(\frac{4}{\kappa} + \frac{8}{3}\right) \log\left(\frac{\lceil 2\log_2 \frac{tK}{2}\rceil t^2}{\delta}\right) + \frac{8}{\kappa} \\
& \le \lambda_{\max}(V_{T_0+1}) + \frac{64}{\kappa^2} d \log\left(\frac{t}{d}\right) + \frac{4}{\kappa}\left(\frac{4}{\kappa} + \frac{8}{3}\right) \log\left(\frac{\lceil 2\log_2 \frac{tK}{2}\rceil t^2}{\delta}\right) + \frac{8}{\kappa} \\
& \le T_0 + \frac{64}{\kappa^2} d \log\left(\frac{t}{d}\right) + \frac{4}{\kappa}\left(\frac{4}{\kappa} + \frac{8}{3}\right) \log\left(\frac{\lceil 2\log_2 \frac{tK}{2}\rceil t^2}{\delta}\right) + \frac{8}{\kappa} + 1
\end{aligned}
$$

where we apply Lemma 10 to bound $\sum_{\tau=T_0+1}^t \max_{i \in S_\tau} \|\widetilde{\boldsymbol{x}}_{\tau i}\|_{V_{\tau+1}^{-1}}^2$.

## D  PROOF OF THEOREM 5

At a high level, we prove Theorem 5 in three steps. In the first step, we construct an adversarial set of parameters and reduce the task of lower bounding the worst-case regret of any policy to lower bounding the Bayes risk over the constructed parameter set. In the second step, we use a counting argument similar to the one used in Chen & Wang (2018) and Chen et al. (2020) to provide an explicit lower bound on the Bayes risk of the constructed adversarial parameter set. Finally, we apply Pinsker's inequality to complete the proof. The following sections provide the details for each of these steps.

Let $\epsilon \in (0, (1 - L_0^2)/d\sqrt{d})$ be a small positive parameter to be specified later. For every subset $W \subseteq [d]$, define the corresponding parameter $\boldsymbol{\psi}_W \in \mathbb{R}^d$ as $[\boldsymbol{\psi}_W]_i = \epsilon$ for all $i \in W$, and $[\boldsymbol{\psi}_W]_i = 0$ for all $i \notin W$. Next, define $\boldsymbol{\phi}^* \in \mathbb{R}^d$ as $[\boldsymbol{\phi}^*]_i = L_0\sqrt{1/d}$ for all $i \in [d]$. Finally, for any $W \subseteq [d]$, define the concatenated parameter vectors $\boldsymbol{\theta}_W \in \mathbb{R}^{2d}$ as $\boldsymbol{\theta}_W = (\boldsymbol{\psi}_W, \boldsymbol{\phi}^*)$. The parameter set that we consider is

$$\boldsymbol{\theta} \in \Theta := \{\boldsymbol{\theta}_W : W \in \mathcal{W}_{d/4}\}$$

where $\mathcal{W}_{d/4} := \{W \subseteq [d] : |W| = d/4\}$ denotes the set of all subsets of $[d]$ whose size is $d/4$. Note that $d/4$ is a positive integer because $d$ is divisible by 4. It is also easy to check that with the condition $\epsilon \in (0, (1 - L_0^2)/\sqrt{d})$, we satisfy $\|\boldsymbol{\theta}\| \leq 1$ for any $\boldsymbol{\theta} \in \Theta$.

The feature vectors $\{\boldsymbol{x}_{ti}\}$ are constructed to be invariant over time iterations $t$. For each $t$ and $U \in \mathcal{W}_{d/4}$, $K$ identical feature vectors $\boldsymbol{x}_U$ are constructed as $[\boldsymbol{x}_U]_i = 2/\sqrt{d}$ for all $i \in U$, and $[\boldsymbol{x}_U]_i = 0$ for all $i \notin U$. Furthermore, it is straightforward to verify that $\|\boldsymbol{x}_U\| \leq 1$ for any $U \in \mathcal{W}_{d/4}$.

Hence, the worst-case regret of any policy $\pi$ can be lower bounded by the worst-case regret of parameters belonging to $\Theta$, which can be further lower bounded by the average regret over a uniform prior over $\Theta$. Formally,

$$\sup_{\boldsymbol{\theta}} \mathbb{E}_{\boldsymbol{x},\boldsymbol{\theta}}^{\pi} \sum_{t=1}^{T} R(S_{\boldsymbol{\theta}}^*, \boldsymbol{p}_{\boldsymbol{\theta}}^*) - R(S_t, \boldsymbol{p}_t) = \max_{\boldsymbol{\theta} \in \Theta} \mathbb{E}_{\boldsymbol{x},\boldsymbol{\theta}}^{\pi} \sum_{t=1}^{T} R(S_{\boldsymbol{\theta}}^*, \boldsymbol{p}_{\boldsymbol{\theta}}^*) - R(S_t, \boldsymbol{p}_t) \tag{19}$$

$$= \frac{1}{|\mathcal{W}_{d/4}|} \sum_{W \in \mathcal{W}_{d/4}} \mathbb{E}_{\boldsymbol{x},\boldsymbol{\theta}_W}^{\pi} R(S_{\boldsymbol{\theta}_W}^*, \boldsymbol{p}_{\boldsymbol{\theta}_W}^*) - R(S_t, \boldsymbol{p}_t) \tag{20}$$

Here, the $R(\cdot)$ function refers to the expected revenue function $R_t(\cdot)$ defined in equation 1. Since both the context vectors and the feature vectors are invariant over time by construction, we drop the time subscript $t$ to simplify the notation. Additionally, $S_{\boldsymbol{\theta}_W}^*$ and $\boldsymbol{p}_{\boldsymbol{\theta}_W}^*$ refer to the optimal size-$K$ assortment and pricing that maximizes expected revenue under the feature parameter $\boldsymbol{\theta}_W$. By construction, it is easy to verify that $S_{\boldsymbol{\theta}_W}^*$ consists of all $K$ items corresponding to feature $\boldsymbol{x}_W$.

For any fixed assortment $S \in \mathcal{S}_K$, let $\boldsymbol{p}^*(S)$ denote the revenue-maximizing price vector to offer with assortment $S$. That is,

$$\boldsymbol{p}^*(S) \in \max_{\boldsymbol{p} \in \mathbb{R}_+^n} R(S, \boldsymbol{p})$$

with entries $p_i^*(S)$. Then, the optimum prices $\boldsymbol{p}_{\boldsymbol{\theta}_W}^* = \boldsymbol{p}^*(S_{\boldsymbol{\theta}_W}^*)$ can be characterized using the following proposition which is a special case of the Proposition 1.

**Proposition 6.** *Consider that items in an assortment $S$ of size $K$ have utility functions $u_i(p) = \alpha_i - \beta_i \cdot p$. Then, the revenue-maximizing prices for offering assortment $S$ are given by*

$$p_i^*(S) = \frac{1}{\beta_i} + B^0(S)$$

*where $B^0(S)$ is the unique fixed point solution $B$ of the equation*

$$B = \sum_{i \in S} \frac{1}{\beta} e^{\alpha_i - \beta_i B - 1}.$$

*Furthermore, the revenue achieved by offering $(S, \boldsymbol{p}^*(S))$ is equal to $B^0(S)$.*

In particular, if all items in an assortment $S$ have the same utility function $u_i(p) = \alpha - \beta \cdot p$, then we can write $B^0(S)$ as the fixed point solution of

$$B = \frac{K}{\beta} e^{\alpha - \beta B - 1}.$$

## D.2 THE COUNTING ARGUMENT

In this section, we derive an explicit lower bound on the Bayes risk in equation 20. For any sequence $\{(S_t, \boldsymbol{p}_t)\}_{t=1}^T$ produced by the policy $\boldsymbol{\pi}$, we first describe an alternative sequence $\{(\widetilde{S}_t, \widetilde{\boldsymbol{p}}_t)\}_{t=1}^T$ that provably enjoys less regret under the feature parameter $\boldsymbol{\theta}_W$.

Let $\{\boldsymbol{x}_{U_1}, \ldots, \boldsymbol{x}_{U_M}\}$ be the set of context vectors of items contained in assortment $S_t$ (if $S_t = \emptyset$, then choose an arbitrary feature vector $\boldsymbol{x}_U$). Let $\widetilde{U}_t$ be the subset among $U_1, \ldots, U_M$ that maximizes $\langle \boldsymbol{x}_{\widetilde{U}_t}, \boldsymbol{\psi}_W \rangle$, where $\boldsymbol{\theta}_W = (\boldsymbol{\psi}_W, \boldsymbol{\phi}^*)$ is the underlying parameter. Let $\widetilde{S}_t$ be the assortment consisting of all $K$ items corresponding to the feature $\boldsymbol{x}_{\widetilde{U}_t}$ and let $\widetilde{\boldsymbol{p}}_t = \boldsymbol{p}^*(\widetilde{S}_t)$ be the optimum prices for assortment $\widetilde{S}_t$ according to Proposition 6. Then, the following lemma holds true.

**Lemma 17.** $R(S_t, \boldsymbol{p}_t) \le R(\widetilde{S}_t, \widetilde{\boldsymbol{p}}_t)$ for feature parameter $\boldsymbol{\theta}_W = (\boldsymbol{\psi}_W, \boldsymbol{\phi}^*)$.

*Proof.* First, from the optimality of prices $\boldsymbol{p}^*(S_t)$ under $S_t$, we have $R(S_t, \boldsymbol{p}_t) \le R(S_t, \boldsymbol{p}^*(S_t))$. Then, by Proposition 6, $R(S_t, \boldsymbol{p}^*(S_t))$ is equal to the unique fixed point solution for

$$B = \sum_{i \in S} \frac{1}{\beta} e^{\alpha_i - \beta_i B - 1}.$$

Note that the expression on the right-hand side of this equation is monotonically increasing in each $\alpha_i$. Therefore, by replacing all $i \in S_t$ with $i \in \widetilde{S}_t$, the $\alpha_i$ values do not decrease and therefore the fixed point does not increase. That is, the fixed-point solution for

$$B = \sum_{i \in \widetilde{S}_t} \frac{1}{\beta} e^{\alpha_i - \beta_i B - 1}. \tag{21}$$

is greater than or equal to $R(S_t, \boldsymbol{p}^*(S_t))$. Since the unique fixed point solution of equation 21 is equal to $R(\widetilde{S}_t, \widetilde{\boldsymbol{p}}_t)$, we have $R(S_t, \boldsymbol{p}^*(S_t)) \le R(\widetilde{S}_t, \widetilde{\boldsymbol{p}}_t)$, completing the proof.

$\square$

To simplify notation, we use $\mathbb{E}_W$ to denote the expectations under parameter $\theta_W$ and policy $\pi$. The following lemma gives a lower bound for $R(S_{\boldsymbol{\theta}_W}^*, \boldsymbol{p}_{\boldsymbol{\theta}_W}^*) - R(\widetilde{S}_t, \widetilde{\boldsymbol{p}}_t)$.

**Lemma 18.** *Suppose* $\epsilon \in (0, 1/d\sqrt{d})$ *and define* $\delta := d/4 - |\widetilde{U}_t \cap W|$. *Then,*

$$R(S_{\boldsymbol{\theta}_W}^*, \boldsymbol{p}_{\boldsymbol{\theta}_W}^*) - R(\widetilde{S}_t, \widetilde{\boldsymbol{p}}_t) \ge \frac{\delta \epsilon}{15 L_0 \sqrt{d}}$$

Define random variables $\widetilde{N}_i := \sum_{t=1}^T \mathbf{1}\{i \in \widetilde{U}_t\}$. Lemma 18 immediately implies

$$\mathbb{E}_W \left[ R(S_{\boldsymbol{\theta}_W}^*, \boldsymbol{p}_{\boldsymbol{\theta}_W}^*) - R(\widetilde{S}_t, \widetilde{\boldsymbol{p}}_t) \right] \ge \frac{\epsilon}{15 L_0 \sqrt{d}} \left( \frac{dT}{4} - \sum_{i \in W} \mathbb{E}_W[\widetilde{N}_i] \right), \forall W \in \mathcal{W}_{d/4}.$$

Summing both sides of this equation over all $W \in \mathcal{W}_{d/4}$ gives

$$\sum_{W \in \mathcal{W}_{d/4}} \mathbb{E}_W \left[ R(S_{\boldsymbol{\theta}_W}^*, \boldsymbol{p}_{\boldsymbol{\theta}_W}^*) - R(\widetilde{S}_t, \widetilde{\boldsymbol{p}}_t) \right] \ge \frac{\epsilon}{15 L_0 \sqrt{d}} \sum_{W \in \mathcal{W}_{d/4}} \left( \frac{dT}{4} - \sum_{i \in W} \mathbb{E}_W[\widetilde{N}_i] \right).$$

Next, we will upper-bound the term $\sum_{W \in \mathcal{W}_{d/4}} \sum_{i \in W} \mathbb{E}_W[\widetilde{N}_i]$. First, define

$$\mathcal{W}_{d/4}^{(i)} := \{ W \in \mathcal{W}_{d/4} : i \in W \}.$$

Then, we swap the order of summation to write

$$\sum_{W \in \mathcal{W}_{d/4}} \sum_{i \in W} \mathbb{E}_W[\widetilde{N}_i] = \sum_{i \in [d]} \sum_{W \in \mathcal{W}_{d/4}^{(i)}} \mathbb{E}_W[\widetilde{N}_i]$$

$$= \sum_{i \in [d]} \sum_{W \in \mathcal{W}_{d/4-1}} \mathbb{E}_{W \cup \{i\}}[\widetilde{N}_i]$$

$$\leq |\mathcal{W}_{d/4-1}| \max_{W \in \mathcal{W}_{d/4-1}} \sum_{i \in [d]} \mathbb{E}_{W \cup \{i\}}[\widetilde{N}_i]$$

$$= |\mathcal{W}_{d/4-1}| \max_{W \in \mathcal{W}_{d/4-1}} \sum_{i \in [d]} \left( \mathbb{E}_W[\widetilde{N}_i] + \mathbb{E}_{W \cup \{i\}}[\widetilde{N}_i] - \mathbb{E}_W[\widetilde{N}_i] \right)$$

$$\leq |\mathcal{W}_{d/4-1}| \left[ \max_{W \in \mathcal{W}_{d/4-1}} \sum_{i \in [d]} \left( \mathbb{E}_{W \cup \{i\}}[\widetilde{N}_i] - \mathbb{E}_W[\widetilde{N}_i] \right) + \frac{dT}{4} \right]$$

where the last step follows from the fact that $\sum_{i \in [d]} \mathbb{E}_W[\widetilde{N}_i] \leq dT/4$ for any fixed $W \in \mathcal{W}_{d/4-1}$.
Next, we note that

$$\frac{|\mathcal{W}_{d/4-1}|}{|\mathcal{W}_{d/4}|} = \frac{\binom{d}{d/4-1}}{\binom{d}{d/4}} = \frac{d/4}{3d/4+1} \leq \frac{1}{3}$$

to write

$$\frac{1}{|\mathcal{W}_{d/4}|} \sum_{W \in \mathcal{W}_{d/4}} \mathbb{E}_W \left[ R(S_{\boldsymbol{\theta}_W}^*, \boldsymbol{p}_{\boldsymbol{\theta}_W}^*) - R(\widetilde{S}_t, \widetilde{\boldsymbol{p}}_t) \right]$$

$$\geq \frac{1}{|\mathcal{W}_{d/4}|} \frac{\epsilon}{15 L_0 \sqrt{d}} \sum_{W \in \mathcal{W}_{d/4}} \left( \frac{dT}{4} - \sum_{i \in W} \mathbb{E}_W[\widetilde{N}_i] \right)$$

$$\geq \frac{\epsilon}{15 L_0 \sqrt{d}} \left( \frac{dT}{4} - \frac{1}{|\mathcal{W}_{d/4}|} \sum_{W \in \mathcal{W}_{d/4}} \sum_{i \in W} \mathbb{E}_W[\widetilde{N}_i] \right)$$

$$\geq \frac{\epsilon}{45 L_0 \sqrt{d}} \left( \frac{dT}{2} - \max_{W \in \mathcal{W}_{d/4-1}} \sum_{i \in [d]} \left| \mathbb{E}_{W \cup \{i\}}[\widetilde{N}_i] - \mathbb{E}_W[\widetilde{N}_i] \right| \right)$$

### D.3 PINSKER'S INEQUALITY

In this section, we upper bound $\left| \mathbb{E}_{W \cup \{i\}}[\widetilde{N}_i] - \mathbb{E}_W[\widetilde{N}_i] \right|$ for any fixed $W \in \mathcal{W}_{d/4-1}$. Let $\mathbb{P}_W$ and $\mathbb{P}_{W \cup \{i\}}$ to denote the probability law under parameter $\theta_W$ and $\theta_{W \cup \{i\}}$, respectively. Then,

$$\left| \mathbb{E}_{W \cup \{i\}}[\widetilde{N}_i] - \mathbb{E}_W[\widetilde{N}_i] \right| \leq \sum_{n=0}^{T} n \cdot \left| \mathbb{P}_W[\widetilde{N}_i = n] - \mathbb{P}_{W \cup \{i\}}[\widetilde{N}_i = n] \right|$$

$$\leq T \cdot \sum_{n=0}^{T} \left| \mathbb{P}_W[\widetilde{N}_i = n] - \mathbb{P}_{W \cup \{i\}}[\widetilde{N}_i = n] \right|$$

$$\leq 2T \cdot \| \mathbb{P}_W - \mathbb{P}_{W \cup \{i\}} \|_{\mathrm{TV}}$$

$$\leq T \sqrt{2 \cdot \mathrm{KL}(\mathbb{P}_W \| \mathbb{P}_{W \cup \{i\}})}$$

where $\|P - Q\|_{\mathrm{TV}} = \sup_A |P(A) - Q(A)|$ is the total variation distance between laws $P$ and $Q$; $\mathrm{KL}(P\|Q) = \int (\log dP/dQ) dP$ is the Kullback-Leibler (KL) divergence between $P$ and $Q$; and the inequality $\|P - Q\|_{\mathrm{TV}} \leq \sqrt{\frac{1}{2} \mathrm{KL}(P\|Q)}$ is the Pinsker's inequality.

Recall that $\{\boldsymbol{x}_{U_1}, \ldots, \boldsymbol{x}_{U_M}\}$ denotes the set of context vectors of items contained in assortment $S_t$. Then, for every $i \in [d]$, define a new random variable $N_i := \frac{1}{K} \sum_{t=1}^{T} \sum_{j=1}^{M} \mathbf{1}\{i \in U_j\}$. The next lemma is used to upper bound the KL divergence term $\mathrm{KL}(\mathbb{P}_W || \mathbb{P}_{W \cup \{i\}})$.

**Lemma 19** (Lemma 6 in Chen et al. (2020)). *For any $W \in \mathcal{W}_{d/4-1}$ and $i \in [d]$,*

$$\mathrm{KL}(\mathbb{P}_W || \mathbb{P}_{W \cup \{i\}}) \leq C_{\mathrm{KL}} \cdot \mathbb{E}_W[N_i] \cdot \epsilon^2/d$$

*for some universal constant $C_{\mathrm{KL}} > 0$.*

Combining Lemma 19 with the final result of the previous subsection, we obtain

$$\frac{1}{|\mathcal{W}_{d/4}|} \sum_{W \in \mathcal{W}_{d/4}} \mathbb{E}_W \left[ R(S^*_{\boldsymbol{\theta}_W}, \boldsymbol{p}^*_{\boldsymbol{\theta}_W}) - R(\widetilde{S}_t, \widetilde{\boldsymbol{p}}_t) \right]$$

$$\geq \frac{\epsilon}{45 L_0 \sqrt{d}} \left( \frac{dT}{2} - T \sum_{i \in [d]} \sqrt{2 C_{\mathrm{KL}} \cdot \mathbb{E}_W[N_i] \cdot \epsilon^2/d} \right)$$

$$\geq \frac{\epsilon}{45 L_0 \sqrt{d}} \left( \frac{dT}{2} - T\epsilon \sqrt{2 C_{\mathrm{KL}} \sum_{i \in [d]} \mathbb{E}_W[N_i]} \right)$$

$$\geq \frac{\epsilon}{45 L_0 \sqrt{d}} \left( \frac{dT}{2} - T\epsilon \sqrt{C'_{\mathrm{KL}} dT} \right)$$

where $C'_{\mathrm{KL}} = C_{\mathrm{KL}}/2$. Setting $\epsilon = \sqrt{d/16 C'_{\mathrm{KL}} T} \in (0, (1 - L_0^2)/d\sqrt{d})$ for sufficiently large $T$, we obtain

$$\sup_{\boldsymbol{\theta}} \mathbb{E}^{\pi}_{\boldsymbol{x}, \boldsymbol{\theta}} \sum_{t=1}^{T} R(S^*_{\boldsymbol{\theta}}, \boldsymbol{p}^*_{\boldsymbol{\theta}}) - R(S_t, \boldsymbol{p}_t) \geq C_0 d\sqrt{T}/L_0$$

for some universal constant $C_0$, completing the proof of the theorem.

### D.4 Proofs for Technical Lemmas

**Lemma 18.** *Suppose $\epsilon \in (0, 1/d\sqrt{d})$ and define $\delta := d/4 - |\widetilde{U}_t \cap W|$. Then,*

$$R(S^*_{\boldsymbol{\theta}_W}, \boldsymbol{p}^*_{\boldsymbol{\theta}_W}) - R(\widetilde{S}_t, \widetilde{\boldsymbol{p}}_t) \geq \frac{\delta \epsilon}{15 L_0 \sqrt{d}}$$

*Proof.* The optimum revenue from offering $K$ identical items with utility functions $u(p) = \alpha - \beta p$ is equal to the unique fixed point solution $B$ of the equation

$$B = \frac{K}{\beta} e^{\alpha - \beta B - 1}. \tag{22}$$

Using the product logarithm function $W(\cdot)$, we can express the optimum revenue as

$$\frac{W(e^{\alpha-1} K)}{\beta} \tag{23}$$

Let $f_K(x) := W(e^{x-1} K)$ and denote its first derivative with $f'_K(x)$ for any $K \geq 1$. Then, by Lemma 20, there exists a constant $C_K < \frac{2}{3} f'_K(0)$ such that

$$f_K(0) + f'_K(0) \cdot x \leq f_K(x) \leq f_K(0) + f'_K(0) \cdot x + C_K \cdot x^2$$

for all $0 \leq x \leq 1$. For the remainder of this proof, let $\boldsymbol{x} = \boldsymbol{x}_W$, $\widetilde{\boldsymbol{x}} = \boldsymbol{x}_{\widetilde{U}_t}$, and $\boldsymbol{\theta} = \boldsymbol{\theta}_W$. Then, we can write

$$R(S^*_{\boldsymbol{\theta}_W}, \boldsymbol{p}^*_{\boldsymbol{\theta}_W}) = f_K(\boldsymbol{x}^\top \boldsymbol{\theta}) \quad \text{and} \quad R(\widetilde{S}_t, \widetilde{\boldsymbol{p}}_t) = f_K(\widetilde{\boldsymbol{x}}^\top \boldsymbol{\theta}).$$

Putting it all together, we can show that

$$R(S_{\boldsymbol{\theta}_W}^*, \boldsymbol{p}_{\boldsymbol{\theta}_W}^*) - R(\widetilde{S}_t, \widetilde{\boldsymbol{p}}_t) \geq \frac{1}{L_0} \left[ (f_K(0) + f_K'(0)\boldsymbol{x}^\top\boldsymbol{\theta}) - \left( f_K(0) + f_K'(0)\widetilde{\boldsymbol{x}}^\top\boldsymbol{\theta} + C_K(\widetilde{\boldsymbol{x}}^\top\boldsymbol{\theta})^2 \right) \right]$$

$$= \frac{1}{L_0} \left[ f_K'(0)(\boldsymbol{x} - \widetilde{\boldsymbol{x}})^\top\boldsymbol{\theta} - C_K(\widetilde{\boldsymbol{x}}^\top\boldsymbol{\theta})^2 \right]$$

$$\geq \frac{f_K'(0)}{L_0} \left[ (\boldsymbol{x} - \widetilde{\boldsymbol{x}})^\top\boldsymbol{\theta} - \frac{2}{3}(\widetilde{\boldsymbol{x}}^\top\boldsymbol{\theta})^2 \right]$$

$$\geq \frac{f_K'(0)}{L_0} \left[ \frac{\delta\epsilon}{\sqrt{d}} - \frac{2d\epsilon^2}{3} \right]$$

$$\geq \frac{f_K'(0)\delta\epsilon}{3L_0\sqrt{d}}$$

where the last three inequalities use the inequality $0 < f''(0) < f_K'(0)$, the definition of $\delta$, and the inequality $d\epsilon^2 \leq \delta\epsilon/\sqrt{d}$ provided that $\epsilon \in (0, 1/d\sqrt{d})$. Lastly, noting that $f_K'(0) > 1/5$ by Lemma 20 for any $K \geq 1$, we conclude the proof. $\qquad\square$

**Lemma 20.** *Let $f_K(x) := W(e^{x-1}K)$ and denote its first derivative with $f_K'(x)$. Then, for any $K \geq 1$,*

*(a) $f_K'(x) > 1/5$ for all $0 \leq x \leq 1$, and*

*(b) there exists a constant $C_K < \frac{2}{3}f_K'(0)$ such that*

$$f_K(0) + f_K'(0) \cdot x \leq f_K(x) \leq f_K(0) + f_K'(0) \cdot x + C_K \cdot x^2$$

*for all $0 \leq x \leq 1$.*

*Proof.* Let $f_K''(x)$ and $f_K^{(3)}(x)$ denote the second and third derivatives of $f_K(x)$ respectively. Using the properties of the product logarithm function, it is easy to show that

$$f_K'(x) = \frac{f_K(x)}{1 + f_K(x)}, \qquad f_K''(x) = \frac{f_K(x)}{(1 + f_K(x))^3}, \qquad f_K^{(3)}(x) = \frac{(1 - 2f_K(x))f_K(x)}{(1 + f_K(x))^5}.$$

For any $K \geq 1$, $f_K(x)$ is a positive and increasing function of $x$. Hence, $\min_{0 \leq x \leq 1} f_K'(x) = f_K'(0)$. Furthermore, we can show that

$$\min_{K \geq 1} f_K'(0) = \min_{K \geq 1} \frac{W(K/e)}{1 + W(K/e)} = \frac{W(1/e)}{1 + W(1/e)} > 1/5$$

proving the first part of the lemma.

To prove the second part of the lemma, we use Taylor's Theorem to write

$$f_K(x) = f_K(0) + f_K'(0) \cdot x + \frac{f_K''(0)}{2} \cdot x^2 + R_K(\zeta; x)$$

$$R_K(\zeta; x) = \frac{f_K^{(3)}(\zeta)}{6} x^3$$

for some $\zeta$ between $0$ and $x$. For any $K \geq 3$, we can easily show that $f_K(x) \geq 1/2$ for all $0 \leq x \leq 1$. Therefore, $R_K(\zeta; x) \leq 0$ for all $0 \leq \zeta \leq x \leq 1$ and we can set $C_K = f_K''(0)/2$ to satisfy the upper bound inequality.

On the other hand, for $K = 1$ and $K = 2$, we can numerically show that

$$\max_{0 \leq \zeta \leq 1} f_K^{(3)}(\zeta) = f_K^{(3)}(0).$$

and $f_K^{(3)}(0) \leq f_K''(0)$. Therefore, we have

$$R_K(\zeta; x) \leq \frac{f_K''(0)}{6} \cdot x^2$$

for all $0 \leq \zeta \leq x \leq 1$ when $K = 1$ or $K = 2$. As a result, we can set $C_K = 2f_K''(0)/3$ to satisfy the upper bound inequality.

Since $f_K''(0) < f_K'(0)$ for any $K \geq 1$, the selected constant $C_K$ also satisfies $C_K < \frac{2}{3} f_K'(0)$.

$\square$

## E  EXPERIMENTAL DETAILS

We numerically evaluate our algorithms over 20 independently generated problem instances and provide our results in Figure 2. We run experiments with $n = 100$ items for various assortment sizes $K$ and various numbers of feature dimensions $d$. In each instance, the parameter $\boldsymbol{\psi}^*$ is uniformly chosen from $\{\boldsymbol{\psi} : \|\boldsymbol{\psi}\|_2 = 1/2\}$. On the other hand, price sensitivity parameter $\boldsymbol{\phi}^*$ is generated by independently drawing its entries from a uniform distribution over $[\sqrt{L_0}/\sqrt{d}, 1/\sqrt{2d}]$ for some parameter $L_0 > 0$. Each context vector $\boldsymbol{x}_{ti}$ is generated by independently drawing its entries over $[\sqrt{L_0}/\sqrt{d}, 1/\sqrt{2d}]$. This construction ensures that we satisfy both Assumptions 2 and 3.

