# OpenReview forum: "Dynamic Assortment Selection and Pricing with Learning"
_ICLR.cc/2024/Conference — Submitted to ICLR 2024_

### Official Review · Reviewer_nk1g · 2023-10-21

**Soundness:** 2 fair
**Presentation:** 3 good
**Contribution:** 2 fair
**Rating:** 3
**Confidence:** 3

**Summary:**

This paper considers the problem of joint pricing and (cardinality-constrained) assortment optimization with contextual information. Specifically, the value of an item of context $x$ and price $p$ is defined as $\langle\psi^\star, x\rangle - \langle\phi^\star, x\rangle\cdot p$ where $\phi^\star$ and $\psi^\star$ are both $d$-dimensional unknown vectors to be learned. Assuming a known lower bound $L_0$ on the price sensitivity $\langle\phi^\star,x\rangle$, the authors first proposed an algorithm (Algorithm 2) achieving optimal $\tilde{\mathcal{O}}(d\sqrt{T})$ regret guarantee for this problem. The algorithm contains two phases. In the first phase, the algorithm generally pick uniformly random sets and prices in order to guarantee lower bounded eigenvalue of the covariance matrix, guaranteeing good bound on the parameter estimation error. In the second phase, the algorithm applies UCB strategy on both the price and the assortment. The authors also provide $\Omega(d\sqrt{T}/L_0)$ regret lower bound for this problem, showing the optimality of their obtained upper bound. Finally, the authors also conducted numerical analysis on their proposed algorithm and showed empirical better performance compared with the ones of the previous algorithms.

**Strengths:**

- The problem considered in this paper is important and the authors first formalized the joint pricing and assortment problem in this contextual bandit setup.

- The proposed algorithms are intuitive and easy to follow. The general idea of the proof is also intuitively correct to me.

- The authors provided both an upper bound of $\tilde{\mathcal{O}}(d\sqrt{T})$ and a lower bound $\Omega(d\sqrt{T})$, showing the near-optimality of their proposed algorithm.

- The authors also conduct experiments for their proposed algorithm, showing the effectiveness and better empirical performance over previous algorithms.

**Weaknesses:**

- One main concern is the novelty in the analysis of the proposed algorithm. While this problem is proposed by the authors and is new, the algorithm design is very similar to the previous work in [Oh & Iyenger 2021]. Specifically, in both works, the algorithm has two phases with the first phase doing uniform exploration on the decision variables to ensure good conditioning on the covariance matrix and the second phase making the decisions based on UCB. While in this work, the decisions involve both the assortment and pricing, given the proposition proposed in [Wang 2013], this UCB step is also not hard to obtain. From the analysis perspective, the proof idea is also similar to [Oh & Iyenger 2021] by showing the consistency and normality of MLE and then the regret bound follows from the standard term decompositions, which in my opinion does not introduce much challenge in the analysis.

- Another concern is about the tightness of the upper bound and the dependency on problem-dependent constants. Specifically, the lower bound has $\Omega(\frac{1}{L_0})$ dependence on the minimum price sensitivity but the dependency on $L_0$ is not explicitly written in the upper bound. With some check, I think $P=\Theta(\frac{1}{L_0})$ (according to Lemma 1), $\kappa = \Theta(\exp(-\frac{1}{L_0}))$ (according to Lemma 3), and $T_0=\Theta(\frac{1}{L_0^2} \exp(\frac{2}{L_0}))$ (according to equation (16)), leading to regret dependent on $\Theta(PT_0)=\Theta(\frac{1}{L_0^3}\exp^{\frac{2}{L_0}})$. This constant can be much larger than the one shown in the lower bound.

- Also the algorithm requires the knowledge of the minimum price sensitivity $L_0$, which is in general not known to the learner in practice.

**Questions:**

- Can the author highlight the main technical challenge in handling both the price decision and the assortment decision? Currently, it seems to me that as long as the uniform exploration phase is done, the remaining analysis follows smoothly from [Oh & Iyenger 2021] and [Wang 2013].

- Can the authors discuss more about the tightness of either the lower bound or the upper bound on problem-dependent constant?

- One question for the analysis: while I understand the general idea of the algorithm and the analysis, for Lemma 6, I wonder how is $|\langle \theta_t,\tilde{x}_t\rangle - \langle\theta^\star, \tilde{x}_t\rangle|\leq (1+p)g_t$ obtained? I suppose that this is due to $||\hat{\theta}_t-\theta^\star|| _{V_t} \leq \alpha_t$ but $||(x_t,-px_t)|| _{M}\leq (1+p)||(x_t,x_t)|| _{M}$ does not hold in general for PSD matrix $M$. Can the authors explain more on the proof of this lemma?

**Details Of Ethics Concerns:**

None.

---

> ### Author Response · Authors · 2023-11-20
>
> 1. Our work considers a scenario with pricing optimization as well as assortment selections. Therefore, the utlitity terms are functions of the corresponding prices rather than being constants as usually considered in the online assortment selection literature. As a result, our algorithm and analysis differ from the previous literature since it must construct functional upper bounds. The main challenge is in this construction since the upper bound should hold true for all price levels.
> 2. We have concentrated on tightness of upper and lower bounds in terms of $T$ and $d$ in our work. We agree that there might be room for improvement in terms of regret dependency to other problem parameters $K$ and $L_0$.
> 3. Regarding your comment on the proof of Lemma 6, we agree that we need to modify the definition of $g_{ti}$ as $ g_{ti} = \alpha_t \max_{1 \leq p \leq P} \||(x_{ti}, -p x_{ti})\||\_{V_t^{-1}}$ . Then, we would obtain $h_{ti}(p) - u_{ti}(p) \leq g_{ti}$ and the given results would still follow.

---

### Official Review · Reviewer_7z3J · 2023-11-11

**Soundness:** 2 fair
**Presentation:** 2 fair
**Contribution:** 2 fair
**Rating:** 3
**Confidence:** 4

**Summary:**

The paper addresses the problem of dynamic assortment selection and pricing in online marketplaces. The authors' approach to this problem is through the development of an algorithm based on the UCB approach, which is applied to a multinomial logit (MNL) choice model​.
The paper claims several advancements. It first formulates the problem of sequential assortment selection and pricing under contextual MNL choice probabilities and proposes a UCB-based algorithm designed to address this issue. The algorithm is notable for achieving an $O(dsqrt{T})$ regret in total $T$ rounds, where $d$ is the dimension of the context vectors, and the regret rate is claimed to be optimal up to logarithmic factors.
The authors then improve the time and space complexity of their algorithm by incorporating online Newton step techniques for parameter estimation.

**Strengths:**

- The paper addresses the problem of dynamic assortment pricing and selection, which is an important problem.
- The paper presents algorithms with regret bounds, both upper and lower bounds, to show the efficiency of the proposed methods.

**Weaknesses:**

- **Dependence on $kappa$**: Dependence on $kappa$ should be stated in the regret bound. The paper does not even state the assumption on  $kappa$ which commonly exists in almost all the existing literature on contextual MNL bandits (Chen et al., 2020; Oh & Iyengar, 2021; Perivier & Goyal, 2022).  I encourage the authors to state the required assumptions clearly in the main text. Rather, the paper shows such dependence on $kappa$ only in the appendix (in Lemma 3). This dependence should be discussed. Particularly, given that the improvement in its dependence in the recent literature such as Perivier & Goyal, 2022 has been already shown.

- **No dependence on $K$ at all?** The previous results (Chen et al., 2020; Oh & Iyengar, 2021; Perivier & Goyal, 2022). have shown logarithmic dependence on $K$. Yet, this paper does not show any dependence on $K$ even logarithmically, which is questionable. How was this independence achieved?

- Comparison with the most recent and related work, Perivier & Goyal (2022), is very limited. Perivier & Goyal (2022) also considered both assortment pricing and selection under the MNL choice model. Although Perivier & Goyal (2022) do not jointly address pricing and assortment selection as this paper did. Yet, there is no sufficient comparison with Perivier & Goyal (2022) both in theory and even in the simulation.

- I believe that Proposition 2 is too obvious to the readers of the contextual MNL bandit literature. There is no need to state this in the limited space of the main text.

- The authors assert that "[they] are the first to address the problem of dynamic contextual assortment selection and pricing." However, there appears to be an existing work by Miao and Chao (link below) that addresses assortment pricing and selection jointly. Miao and Chao's work is in a non-contextual setting. Yet, the authors at least need to cite their work and provide a comparison.

Miao and Chao: https://papers.ssrn.com/sol3/papers.cfm?abstract_id=3173267

**Questions:**

- Please address the points and questions provided in the weaknesses.
- The regret analysis under MNL model has been studied quite extensively in recent years. Also, online parameter estimation techniques are well-known. This paper heavily depends on the previous techniques, which could be fine as long as the paper bring at least some newer insights. Yet, I really wonder what technical advancements this paper provides. Can you elaborate on what are main technical challenges that could not be solved by simply applying previous techniques?
- Context stochasticity in Assumption 3. If the stochasticity is only required during the initialization period, then authors may consider just regularization as did in many of the previous literature. Why do you stick with Assumption 3? What happens when you do not have this assumption but use regularized MLE instead?

---

> ### Author Response · Authors · 2023-11-20
>
> 1. In our work, we do not directly make an assumption of the $\kappa$ value. However, our assumption regarding the $L_0$ parameter directly translates into Lemma 3, which states that there is a sufficiently large probability (i.e., at least $\kappa$ probability) of selection for any item $i \in S_t$ under any parameter $\boldsymbol{\theta}$ sufficiently close to the true parameter $\boldsymbol{\theta}^*$.
> 2. As stated above Theorem 3, our regret upper bound does not capture the dependency with respect to the assortment size parameter K. Since the maximum assortment size is typically small (i.e., K = O(1)) in many real-world applications, we concentrate on the dependency to $T$ and $d$ as an initial step in obtaining matching upper and lower regret upper bounds.
> 3. Our work considers a scenario with pricing optimization as well as assortment selections. Therefore, the utlitity terms are functions of the corresponding prices rather than being constants as usually considered in the online assortment selection literature. As a result, our algorithm and analysis differ from the previous literature since it must construct functional upper bounds.
> 4. Thank you for pointing out the existing work by Miao and Chao that addresses assortment pricing and selection jointly in a non-contextual setting. We will cite their work and provide a comparison. Similarly, we will elaborate more on the comparison with Perivier & Goyal (2022).

---

> > ### Comment · Reviewer_7z3J · 2023-11-23
> >
> > Thanks for your responses. Here are suggestions for your future submission.
> >
> > 1. That is precisely the reason why the assumptions need to be stated clearly. If you are not going to use an assumption, then this is a point connected to how you present $K$ in the second point below. Lemma 3 shows that in $\kappa$ is $O(1/K^2)$. You need to clearly state these kinds of dependence clearly.
> > 2. You can refer to many existing literature on MNL bandits and dynamic assortments. Dependence on $K$ is clearly stated. $K$ is not something you can sweep under the rug... Please state the $K$ dependence.

---

### Official Review · Reviewer_zWYm · 2023-11-11

**Soundness:** 3 good
**Presentation:** 2 fair
**Contribution:** 1 poor
**Rating:** 3
**Confidence:** 4

**Summary:**

This article introduces an algorithm based on Upper Confidence Bound (UCB) for tackling the problem of online assortment selection and pricing optimization with observable: the utility of each item is a linear function of these features (and price) but the exact coefficients are unknown, and the customer’s choice will follow an MNL model. The paper also provides the upper regret bound of the UCB algorithm which matches the offered lower regret bound of this problem.

**Strengths:**

The paper is well-written, addresses a practical problem, and offers a UCB-based algorithm with theoretical guarantees that matches the lower bound.

**Weaknesses:**

1) the assortment optimization in (5) and Line 5 in Algorithm 1 both require finding an assortment that maximizes revenue under given prices. This doesn't seem trivial to me, and I don't know whether this is even an NP-hard problem. More discussions on this is necessary.

(2) The main algorithm (Algorithm 2) requires initialization rounds involving uniformly random selection of assortments and prices, which could be a practical challenge. Such fluctuations in prices and assortments can be undesirable in practice. As noted in the proof sketch of Theorem 3, these rounds just aim to ensure the invertibility of the design matrix. Then, would an L2-regularized/ridge-type MLE objective solve such an invertibility issue?

(3) The contribution of this paper is mainly on the modeling part, while the technical contribution is limited. The proof skills are standard. The author should highlight the challenge in the proof and the technical contribution.

(4) Relevance. I am not sure if the paper and the topic studied would be of general interest to ICLR audience.

**Questions:**

See above.

---

> ### Author Response · Authors · 2023-11-20
>
> 1. Each step of algorithm 1 requires computing $v_{ti}(B)$ value for each item $i$ and finding the top-$k$ among these numbers. Therefore, each step has a computational complexity of $O(n)$ as described in Section 3. On the other hand, since Algorithm 1 is running a binary search to find the fixed point $B$, it only needs $\log(1/\epsilon)$ iterations to reach $\epsilon$ accuracy. For some constant $\epsilon$ of our choice, the number of iterations also become a small consant.
> 2. We agree on that initialization rounds in Algorithm 2 lead to fluctuations in the prices. As you noted out, one solution approach might be to consider a regularized MLE problem. However, it will lead to major changes in the analysis of the algorithm and therefore it is out of scope of our current work.
> 3. Our work considers a scenario with pricing optimization as well as assortment selections. Therefore, the utlitity terms are functions of the corresponding prices rather than being constants as usually considered in the online assortment selection literature. As a result, our algorithm and analysis differs from the previous literature since it must construct functional upper bounds.

---

### Official Review · Reviewer_5Wv4 · 2023-11-12

**Soundness:** 2 fair
**Presentation:** 2 fair
**Contribution:** 2 fair
**Rating:** 3
**Confidence:** 3

**Summary:**

The paper studies an online assortment and pricing problem, where customers arrive online and make decisions under an MNL model. To solve this contextual MNL bandit problem, the authors developed a UCB algorithm, achieving an O(\sqrt{T}) regret. The numerical experiments also show the performance of the algorithm.

**Strengths:**

1. The paper is written clearly.
2. The problem looks new.
3. The algorithm performs good theoretically and numerically.

**Weaknesses:**

1. It looks like the idea of this algorithm and the corresponding analysis are standard.
2. It seems Algorithm 1 is very computationally expensive. Could the authors have more discussion of the running time of Algorithm 1?
3. The assumptions look strong. For example, $L_0$ is not observable in practice. I wonder whether the algorithm or a modified algorithm still works if $L_0$ is unknown.
4. The choice of S_t and p_t in Algorithm 2 looks inconsistent with the proof. The choice in the algorithm is based on Algorithm 1, which is an estimated algorithm, while the proof assumes the chosen assortment and prices are optimal. I wonder whether the authors can elaborate more on this inconsistency.

**Questions:**

See weaknesses

---

> ### Author Response · Authors · 2023-11-20
>
> 1. Our work considers a scenario with pricing optimization as well as assortment selections. Therefore, the utlitity terms are functions of the corresponding prices rather than being constants as usually considered in the literature. As a result, our algorithm and analysis differs from the previous literature since it must construct functional upper bounds.
> 2. Each step of algorithm 1 requires computing $v_{ti}(B)$ value for each item $i$ and finding the top-$k$ among these numbers. Therefore, each step has a computational complexity of $O(n)$ as described in Section 3. On the other hand, since Algorithm 1 is running a binary search to find the fixed point $B$, it only needs $\log(1/\epsilon)$ iterations to reach $\epsilon$ accuracy. For some constant $\epsilon$ of our choice, the number of iterations also become a small consant.
> 3. The assumption regarding $L_0$ directly translates into Lemma 3, which states that there is a sufficiently large probability (i.e., at least $\kappa$ probability) of selection for any item $i \in S_t$ under any parameter $\boldsymbol{\theta}$ sufficiently close to the true parameter $\boldsymbol{\theta}^*$. The statement of this lemma is usually assumed to be true as an assumption in the literature of MNL bandits (Oh and Iyengar 2021) and the algorithms depend on the value $\kappa$. Following a similar approach here, we make use of our assumption on the $L_0$ parameter. Even though it might be posssible to estimate (or estimate a lower bound for) the parameter $L_0$ using choice observations, it is out of scope of our work here.
> 4. Algorithm 1 is described for utility parameters $\alpha_{ti}$ and $\beta_{ti}$ given as inputs to the algorithm. In Algorithm 2, we call Algorithm 1 according to linear utility functions $h_{ti}(p)$ (i.e., we run Algorithm 1 according to the coefficients of each h_{ti}(p) function).

---

### Author Response · Authors · 2023-11-20

We thank all the reviewers for their constructive feedback and thoughtful questions. Please see our separate responses below each review.

---

### Meta-Review · Area_Chair_p99v · 2023-12-09

**Metareview:**

The algorithm proposed in the paper is quite similar to [Oh & Iyenger 2021], and the analysis is standard. The novelty of the paper -- besides introducing the problem of dynamic assortment selection and pricing -- is limited in the presence of existing literature of the dynamic pricing work. All reviewers have serious concerns about the contribution of the paper, which the paper does not make clear. I share a similar a view and believe the paper is not publishable as it currently stands.

**Justification For Why Not Higher Score:**

N/A

**Justification For Why Not Lower Score:**

N/A

---

### Decision · Program_Chairs · 2024-01-16

Reject